# TECRL, a new life-threatening inherited arrhythmia gene associated with overlapping clinical features of both LQTS and CPVT

Harsha D Devalla[1,*,†], Roselle Gélinas[2,3,†], Elhadi H Aburawi[4,†], Abdelaziz Beqqali[5,†], Philippe Goyette[2], Christian Freund[1,6], Marie-A Chaix[2,3], Rafik Tadros[2,3,5], Hui Jiang[7,8,9], Antony Le Béchec[10], Jantine J Monshouwer-Kloots[1], Tom Zwetsloot[1], Georgios Kosmidis[1], Frédéric Latour[2], Azadeh Alikashani[2], Maaike Hoekstra[5], Jurg Schlaepfer[11], Christine L Mummery[1], Brian Stevenson[10], Zoltan Kutalik[10,12], Antoine AF de Vries[13,14], Léna Rivard[2,3], Arthur AM Wilde[15,16], Mario Talajic[2,3], Arie O Verkerk[5,‡], Lihadh Al-Gazali[4,‡], John D Rioux[2,3,**,†,‡], Zahurul A Bhuiyan[17,***,‡] & Robert Passier[1,18,****,‡]

## Abstract

Genetic causes of many familial arrhythmia syndromes remain elusive. In this study, whole-exome sequencing (WES) was carried out on patients from three different families that presented with life-threatening arrhythmias and high risk of sudden cardiac death (SCD). Two French Canadian probands carried identical homozygous rare variant in *TECRL* gene (p.Arg196Gln), which encodes the *trans*-2,3-enoyl-CoA reductase-like protein. Both patients had cardiac arrest, stress-induced atrial and ventricular tachycardia, and QT prolongation on adrenergic stimulation. A third patient from a consanguineous Sudanese family diagnosed with catecholaminergic polymorphic ventricular tachycardia (CPVT) had a homozygous splice site mutation (c.331+1G>A) in *TECRL*. Analysis of intracellular calcium ($[Ca^{2+}]_i$) dynamics in human induced pluripotent stem cell-derived cardiomyocytes (hiPSC-CMs) generated from this individual ($TECRL_{Hom}$-hiPSCs), his heterozygous but clinically asymptomatic father ($TECRL_{Het}$-hiPSCs), and a healthy individual (CTRL-hiPSCs) from the same Sudanese family, revealed smaller $[Ca^{2+}]_i$ transient amplitudes as well as elevated diastolic $[Ca^{2+}]_i$ in $TECRL_{Hom}$-hiPSC-CMs compared with CTRL-hiPSC-CMs. The $[Ca^{2+}]_i$ transient also rose markedly slower and contained lower sarcoplasmic reticulum (SR) calcium stores, evidenced by the decreased magnitude of caffeine-induced $[Ca^{2+}]_i$ transients. In addition, the decay phase of the $[Ca^{2+}]_i$ transient was slower in $TECRL_{Hom}$-hiPSC-CMs due to decreased SERCA and NCX activities. Furthermore, $TECRL_{Hom}$-hiPSC-CMs showed prolonged action potentials (APs) compared with CTRL-hiPSC-CMs. *TECRL* knock-down in control human embryonic stem cell-derived CMs (hESC-CMs) also resulted in significantly longer APs. Moreover, stimulation by noradrenaline (NA) significantly increased the

---

1   Department of Anatomy & Embryology, Leiden University Medical Center, Leiden, The Netherlands
2   Montreal Heart Institute, Montreal, QC, Canada
3   Department of Medicine, Université de Montréal, Montreal, QC, Canada
4   Department of Pediatrics, College of Medicine and Health Sciences, UAE University, Al Ain, United Arab Emirates
5   Heart Failure Research Center, Academic Medical Center, University of Amsterdam, Amsterdam, The Netherlands
6   Leiden University Medical Center hiPSC Core Facility, Leiden, The Netherlands
7   Beijing Genomics Institute, Shenzhen, China
8   Shenzhen Key Laboratory of Genomics, Shenzhen, China
9   The Guangdong Enterprise Key Laboratory of Human Disease Genomics, Shenzhen, China
10  Vital-IT group, Swiss Institute of Bioinformatics, Lausanne, Switzerland
11  Service de Cardiologie, Centre Hospitalier Universitaire Vaudois (CHUV), Lausanne, Switzerland
12  Institute of Social and Preventive Medicine, University Hospital (CHUV) and University of Lausanne, Lausanne, Switzerland
13  Laboratory of Experimental Cardiology, Department of Cardiology, Leiden University Medical Center, Leiden, The Netherlands
14  ICIN-Netherlands Heart Institute, Utrecht, The Netherlands
15  Heart Center, Department of Clinical and Experimental Cardiology, Academic Medical Center, University of Amsterdam, Amsterdam, The Netherlands
16  Princess Al-Jawhara Al-Brahim Centre of Excellence in Research of Hereditary Disorders, Jeddah, Saudi Arabia
17  Laboratoire Génétiqué Moléculaire, Centre Hospitalier Universitaire Vaudois (CHUV), Lausanne, Switzerland
18  Department of Applied Stem Cell Technologies, MIRA Institute for Biomedical Technology and Technical Medicine, University of Twente, Enschede, The Netherlands
    *Corresponding author. Tel: +31 715268889; E-mail: h.d.devalla@lumc.nl
    **Corresponding author. Tel: +1 5143763330 ext. 3741; E-mail: john.david.rioux@umontreal.ca
    ***Corresponding author. Tel: +41 213143370; E-mail: z.a.bhuiyan@chuv.ch
    ****Corresponding author. Tel: +31 534895553; E-mail: r.passier@lumc.nl
    †These authors contributed equally to this work
    ‡These authors contributed equally to this work

 

propensity for triggered activity based on delayed afterdepolarizations (DADs) in TECRL$_{Hom}$-hiPSC-CMs and treatment with flecainide, a class Ic antiarrhythmic drug, significantly reduced the triggered activity in these cells. In summary, we report that mutations in *TECRL* are associated with inherited arrhythmias characterized by clinical features of both LQTS and CPVT. Patient-specific hiPSC-CMs recapitulated salient features of the clinical phenotype and provide a platform for drug screening evidenced by initial identification of flecainide as a potential therapeutic. These findings have implications for diagnosis and treatment of inherited cardiac arrhythmias.

**Keywords** Arrhythmia; CPVT; iPSC; LQTS; SRD5A2L2
**Subject Categories** Cardiovascular System; Genetics, Gene Therapy & Genetic Disease

See also: **MD Perry & JI Vandenberg** (December 2016)

## Introduction

Inherited arrhythmogenic diseases (IADs) are one of the prevalent causes of sudden cardiac death (SCD) in the young (Wilde & Behr, 2013). IADs can be classified into disorders with or without structural heart defects. The latter include channelopathies such as long QT syndrome (LQTS) and catecholaminergic polymorphic ventricular tachycardia (CPVT), caused by mutations in genes encoding ion channel or calcium-handling proteins that primarily affect the electrical activity of the heart (Schwartz *et al*, 2013; Wilde & Behr, 2013).

LQTS is most commonly inherited in an autosomal dominant mode, where mutations in *KCNH2*, *KCNQ1*, and *SCN5A* account for the majority of cases (Wilde & Behr, 2013). A very rare autosomal recessive form of LQTS, often accompanied with sensorineural deafness (Jervell–Lange-Nielsen syndrome), has been linked to mutations in *KCNQ1* and *KCNE1*. At least 12 other genes have been linked to LQTS. However, 20–30% of LQTS cases remain genetically elusive. The mechanism of arrhythmia in LQTS mainly involves QT prolongation with early afterdepolarization-mediated triggered activity, which can lead to *torsades de pointes* (TdP) and ventricular fibrillation.

CPVT is also commonly inherited as an autosomal dominant disorder due to mutations in the cardiac ryanodine receptor, *RYR2* (65% of all cases). Rare autosomal recessive mutations in the calcium-sequestering protein, *CASQ2* (2–5% of all cases), account for a small fraction of CPVT population (Lahat *et al*, 2001; Laitinen *et al*, 2001; Priori *et al*, 2001; Postma *et al*, 2002). Mutations in *TRDN* (Roux-Buisson *et al*, 2012), a calcium release complex protein, and *CALM1* (Nyegaard *et al*, 2012), a calcium-binding protein, have also been implicated in CPVT. Interestingly, *CALM1*, *CALM2*, and *CALM3* mutations have also been linked to early-onset LQTS (Crotti *et al*, 2013; Reed *et al*, 2015; Chaix *et al*, 2016), highlighting the locus-disease heterogeneity in IADs. Clinically, a normal electrocardiogram (ECG) at rest and typical arrhythmias, including

bi-directional and polymorphic ventricular tachycardia (VT), in response to catecholaminergic stress is observed in CPVT. Susceptibility to VT in response to β-adrenergic stimulation is the hallmark of CPVT, which manifests as exercise-induced syncope and sudden death (Leenhardt *et al*, 2012).

Although understanding of genetic loci associated with IADs has improved remarkably over the last few years, in a number of cases, the genetic basis of the clinical phenotype has remained elusive (Schwartz *et al*, 2013). Genetic testing in LQTS and CPVT has been broadly adopted as a screening tool for these potentially fatal diseases. In cases without known disease-causing mutations, identifying family members at risk of SCD can however be challenging. Therefore, discovering novel Mendelian disease-causing loci through genetic testing has the potential to improve clinical care and prevent SCD.

In this study, we identified mutations in *trans*-2,3-enoyl-CoA reductase-like (*TECRL*) gene in three patients with clinical arrhythmias in whom none of the mutations in the most common LQTS and CPVT genes had been detected. Two of these patients were of French Canadian origin but unrelated and were diagnosed with LQTS at the Cardiovascular Genetics Center of the Montreal Heart Institute (MHI). They had strikingly similar cardiac electrical phenotypes characterized by normal QT interval on their ECG at baseline, adrenergic-induced QT prolongation as well as atrial and ventricular arrhythmias including aborted SCD. Whole-exome sequencing (WES) of these patients revealed the presence of an identical homozygous rare variant in exon 6 of *TECRL*, also annotated *SRD5A2L2* (p.Arg196Gln).

A third patient included in this study belongs to a consanguineous Arab family of Sudanese origin reported previously (Bhuiyan *et al*, 2007). Members of this family were diagnosed with early-onset and highly malignant form of CPVT and segregation analysis indicated an autosomal recessive mode of inheritance. Sequencing of the coding exons and exon–intron boundaries of *RYR2* and *CASQ2* and several other cardiac genes such as *KCNJ2*, *FKBP12.6*, *SCN5A*, *KCNH2*, *KCNQ1*, *KCNE1*, *KCNE2*, and *NCX1* had not revealed any mutations. Here, we identified a homozygous G>A point mutation in the splice donor site of intron 3 of *TECRL* (*TECRL*c.331 + 1G > A) in the affected individuals of this family. To date, all but two children, who inherited this mutation in the affected family, died following a cardiac event during physical activity. To understand the functional consequence of the *TECRL*c.331 + 1G > A mutation and model the disease phenotype *in vitro*, we generated human induced pluripotent stem cells (hiPSCs) (Takahashi *et al*, 2007) from a 5-year-old symptomatic patient (referred to as TECRL$_{Hom}$-hiPSCs), his heterozygous (but clinically asymptomatic) father (TECRL$_{Het}$-hiPSCs), and a family member free of the mutation (CTRL-hiPSCs). hiPSCs were differentiated into cardiomyocytes (CMs) and analyzed *in vitro*. Using the patient-derived hiPSC-CMs, we showed that the c.331 + 1G > A mutation in *TECRL* leads to skipping of exon 3. TECRL$_{Hom}$-hiPSC-CMs recapitulated aspects of the disease phenotype *in vitro* including increased susceptibility to triggered activity, which could be alleviated by treatment with flecainide.

Taken together, the clinical, genetic, and experimental results from this study have identified *TECRL* as a new gene associated with life-threatening inherited arrhythmias displaying features of both LQTS and CPVT.

# Results

### Clinical data

This study reports three patients from three different families presenting clinically with life-threatening arrhythmias and cardiac arrest followed by successful resuscitation. Two of these patients were diagnosed with LQTS at the Cardiovascular Genetics Center of the Montreal Heart Institute following investigation for aborted cardiac arrest. They had distinctive clinical features of recurrent exercise- and emotion-induced atrial and ventricular arrhythmias. The third patient was from a large consanguineous family with two sub-families and several children affected with adrenergic-related lethal events and were previously diagnosed with CPVT (Bhuiyan *et al*, 2007). Detailed clinical manifestations of these three patients and additional family members are described below and in the Appendix.

Patient 1 is a French Canadian female, who presented with ventricular fibrillation and cardiac arrest during walking at age 22, without prior history of syncope or documented arrhythmia. Coronary angiography, transthoracic echocardiography, and invasive electrophysiological (EP) testing, including ventricular premature stimulation, were normal. The resting ECG showed a normal QT interval, but isoproterenol infusion resulted in QT prolongation. During follow-up, the patient presented with recurrent episodes of exercise- or emotion-induced atrial and ventricular arrhythmias resulting in multiple shocks from an implantable cardioverter–defibrillator (ICD). Arrhythmia was refractory to standard beta-blockers (metoprolol and bisoprolol) but was finally suppressed by high doses of nadolol. During further follow-up, repeat echocardiography remained normal but significant repolarization abnormalities with QT prolongation (QTc range: 451–494 ms) during follow-up were occasionally seen on the resting ECG (Fig 1A); the patient was therefore diagnosed with LQTS. The parents were not known to be consanguineous but came from the same town. No family member was available for genetic testing.

Patient 2 is also a French Canadian female. She presented initially at age 18 with numerous syncopal episodes triggered by emotional stress, with documentation of non-sustained VT during exercise stress testing. Despite beta-blocker therapy, she had an aborted cardiac arrest during emotional stress at the age of 31. At rest, the QTc was 437 ms with repolarization abnormalities (Fig 1B). Coronary angiography and cardiac magnetic resonance imaging were unremarkable. Sustained polymorphic VT was inducible with one ventricular extrastimulus during isoproterenol infusion. Epinephrine challenge resulted in a 57 ms paradoxical QT prolongation with appearance of ventricular bigeminy, suggesting a low repolarization reserve compatible with LQTS (Fig 1C). During follow-up, the patient had multiple episodes of adrenergic atrial and ventricular arrhythmia resulting in numerous ICD shocks. Electrophysiological mapping showed an extensive low-voltage area along the interatrial septum and multiple foci of atrial tachycardia (AT) was ablated. The combination of AT ablation and nadolol significantly decreased arrhythmia recurrence in this patient.

Patient 3 (Subject 1V:13 of Fig 1D) is a Sudanese male who presented with cardiac arrest at age 4 while running. ECG recording during successful resuscitation showed ventricular fibrillation and TdP, which was reversed to sinus rhythm following DC shock. ECG

showed a QTc interval of 450 ms (Fig 1E). He had a second attack while in hospital but was in sinus rhythm between the two attacks with a normal QRS axis, aQTc interval of 450 ms, and no ST changes. He later received an ICD and has had no further syncope or cardiac arrest. However, ICD interrogation occasionally revealed a fast rhythm around 193 beats/min (cycle length 310 ms) with polymorphic ventricular ectopy (Fig 1F). The parents of patient 3 are first-degree cousins and seven of 13 children in the family presented exertion-induced arrhythmias and/or SCD during early childhood. In five of these children, an arrhythmic episode was fatal. Two children (IV:9 and IV:13) survived an arrhythmic attack. Clinical features of the other affected subjects are summarized in Appendix Table S1. IV:1, IV:2, IV:4, and IV:10 have been described earlier (Bhuiyan *et al*, 2007). Clinical details of IV:8, IV:9, and IV:13 are further detailed in the Appendix.

In summary, the clinical phenotype of the affected individuals is characterized by: (i) adrenergic VTs with high prevalence of cardiac arrest and SCD, (ii) recurrent AT sometimes triggering ventricular arrhythmias, and (iii) normal or mildly prolonged QTc at baseline with paradoxical QT increase during adrenergic stimulation. Taken together, the patients from all three families presented a similar severe arrhythmia phenotype with overlapping features of both LQTS and CPVT.

### WES revealed mutations in *TECRL* in all three patients

To uncover the underlying genetic cause of arrhythmias in these patients, WES was performed on genomic DNA from patients and family members when available. Given the low prevalence of IADs, the fact that LQTS or CPVT can independently be caused by many different genes, and reasoning that most mutations identified to date are non-synonymous and familial in nature, we elected to focus our analyses exclusively on novel non-synonymous variants.

#### WES revealed an identical homozygous missense mutation in TECRL in the French Canadian patients

Patient 1 and patient 2 underwent clinical genetic testing and did not contain mutations in *KCNQ1, KCNH2, SCN5A, KCNE1,* or *KCNE2,* five genes most frequently implicated in LQTS. Overall, WES identified 57,828 high-quality single nucleotide variants (SNVs) and in/dels for these two subjects, 231 of which were novel non-synonymous coding and splice site variants. Given the striking similarity in disease phenotype, patient 1 and patient 2 were screened for variants in the same gene, which resulted in the identification of an identical novel homozygous single base pair (bp) mutation in *TECRL*, resulting in an arginine to glutamine substitution at position 196 (p.Arg196Gln). We developed a genotyping assay and confirmed independently that the mutation was homozygous in the two patients and absent in 540 European-derived chromosomes. The p.Arg196Gln substitution is predicted to be "probably damaging" by PolyPhen-2, deleterious by SIFT, and is within a site with a high Genomic Evolutionary Rate Profiling (GERP) score (5.11).

#### WES revealed a common homozygous splice site mutation in TECRL in the affected members of the Sudanese family

The pedigree of the Sudanese family (Fig 1D) was compatible with autosomal recessive inheritance. To reveal the underlying genetic

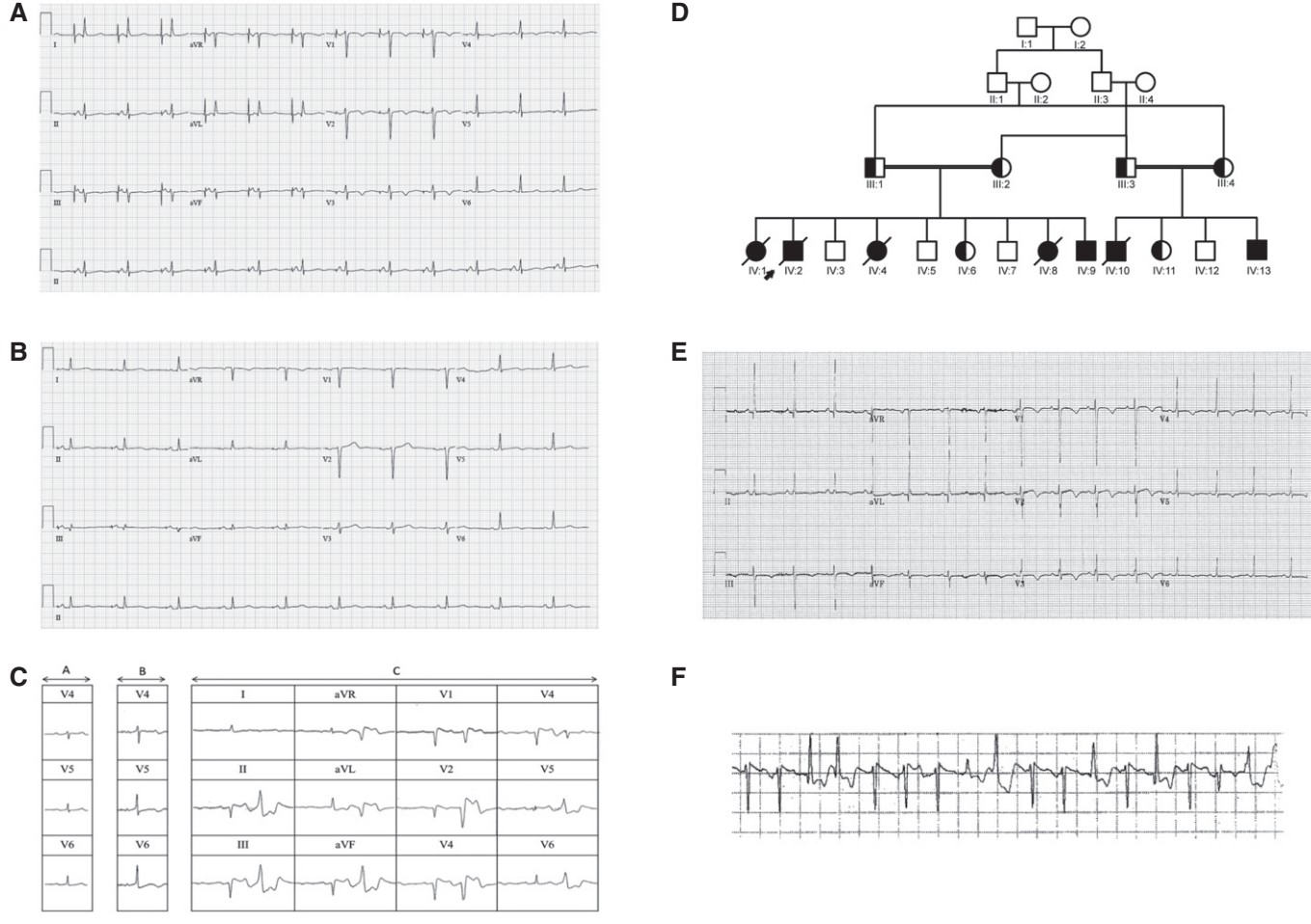

**Figure 1. Arrhythmias in three different patients.**

A  12-lead ECG of patient 1.

B  12-lead ECG of patient 2.

C  ECG of patient 2 during epinephrine test [A, baseline ECG; B, ECG at 17 min and 47 s, at the beginning of epinephrine infusion (0.2 μg/kg of epinephrine); C, ECG after the end of epinephrine infusion].

D  Pedigree of a family diagnosed with arrhythmias; arrow indicates the proband, and solid symbols represent family members affected by arrhythmias who are also homozygous for mutations identified by exome sequencing. Crossed symbols indicate deceased individuals. Half-filled symbols correspond to individuals heterozygous for mutations identified by exome sequencing.

E  ECG of patient 3 (subject IV:13) showing mild QTc prolongation.

F  ICD interrogation of patient 3 (subject IV:13) reveals an episode of VT.

cause of SCD, WES was performed on genomic DNA from two children, IV:2 and IV:10, who are first-degree cousins (Fig 1D) with clinical symptoms. Genomic DNA from the parents of IV:2 (III:1 and III:2; Fig 1D), who are clinically normal, was also included for WES. On average, this yielded ≥ 81.5 million reads per sample, 87% of which could be mapped. The mean coverage of the target region was < 103-fold, with over 93% of target regions covered by ≥ 10 reads. In total 67,000–78,000 SNVs and 4,300–5,400 small in/dels were identified in each of the individuals, of which 1,401–1,789 were novel non-synonymous coding and splice site variants.

We then prioritized variants according to disease inheritance pattern in the index patient, his parents, and affected cousin. This resulted in identification of five SNVs (Appendix Table S2), all shared by IV:2 and IV:10 as homozygous and III:1 and III:2 as

heterozygous following expected disease inheritance pattern. All these variants were further tested in the family members by Sanger sequencing. Of these, a single bp mutation in *ANKRD17* (p.Val1368Gly) and a splice site mutation in *TECRL* (c.331+1G>A) were found to be homozygous in all children who had presented with a clinical phenotype (IV:2, IV:4, IV:8, IV:9, IV:10, and IV:13; Fig 1D) while the parents (III:1, III:2, III:3, and III:4) were found to be heterozygous in both families. *ANKRD17* c.4103T>G and *TECRL* c.331+1G>A variants were not reported as single nucleotide polymorphisms (SNPs) in the general population (neither in dbSNP, 1000 Genomes, 6500 NHLBI, ExAC browser nor in our in-house CoLaus and Vital-IT/SIB databases) and indicated a substitution deficit and evolutionary conservation (phastCons-46 way score of 0.498 and GERP$^{++}$ score of 5.32 for *ANKRD17*; phastCons-46 way

score of 0.397 and GERP$^{++}$ score of 5.23 for *TECRL*). Additionally, these mutations were not found in any of control DNA samples obtained from 72 Saudi Arabian individuals. Importantly, no *ANKRD17* variant was identified in the two French Canadian patients.

Taken together, the results from the French Canadian and Sudanese patients suggested that mutations in *TECRL* are the most likely cause of their life-threatening arrhythmia.

### *TECRL* is an endoplasmic reticulum (ER) protein expressed preferentially in the heart

Human *TECRL* maps to chromosome 4q13 and contains 12 exons (Appendix Fig S1A). The open reading frame of *TECRL* encodes a 363 amino acid protein and predicted to contain a ubiquitin-like domain in the N-terminal half of the protein, three transmembrane segments as well as a 3-oxo-5-alpha steroid 4-dehydrogenase domain in the C-terminal half of the protein (Appendix Fig S1B and C). The homozygous mutation in *TECRL* found in patients 1 and 2 (p.Arg196Gln), is located in 6th exon of *TECRL*, and causes an amino acid substitution immediately upstream of the 3-oxo-5-alphasteroid 4-dehydrogenase domain (Appendix Fig S1C). On the other hand, *TECRL*c.331+1G>A mutation found in patient 3 is located in the splice donor site of intron 3 resulting in an internal deletion in the putative ubiquitin-like domain (Appendix Fig S1A–C). Comparison of amino acid sequences of *TECRL* revealed a high degree of cross-species conservation (Appendix Fig S1D), suggesting that *TECRL* might have an essential function in the heart.

Previously, *TECRL* was identified in a genome-wide transcriptional profiling study of human embryonic stem cells (hESCs) differentiating to CMs *in vitro*. Moreover, in the mouse embryo, its expression was restricted to the heart and inflow tract (Beqqali *et al*, 2006). To further define the spatiotemporal expression pattern of *Tecrl* during embryonic development, its expression in early mouse embryos was investigated by *in situ* hybridization. *Tecrl* was not expressed in the cardiac crescent at embryonic day E7.5 (Fig 2A) but was observed at E8.5 in the entire heart with the strongest expression occurring in the developing inflow tract (Fig 2B), especially in the left horn (Fig 2C). At E9.5, *Tecrl* expression was still detectable in the atria and ventricles, albeit at lower levels whereas strong expression remained in the inflow tract (Fig 2D). From E10 onwards, *Tecrl* was also expressed at low levels in somites, particularly in the myotome region, that gives rise to skeletal muscle (Fig 2E–F). At E10.5, cardiac expression of *Tecrl* was no longer restricted to the inflow tract and its expression was maintained in the somites (Fig 2G). At E14.5, *Tecrl* was expressed in the entire myocardium but not in the lungs, which served as negative control (Fig 2H). In adult mice, reverse transcription quantitative polymerase chain reaction (RT–qPCR) analyses showed that *Tecrl* expression was highest in the heart with very low to almost undetectable levels in brain, skeletal muscle, stomach, pancreas, liver, kidney, small intestine, and uterus (Appendix Fig S1E).

Next, we investigated the distribution of *TECRL* in human tissues. Interestingly, *TECRL* has important sequence identity with *TECR*, a gene that encodes a multi-pass membrane protein that resides in the ER, and belongs to the steroid 5-alpha reductase family (Moon & Horton, 2003). Expression analysis of *TECR* and

*TECRL* across a panel of human tissues clearly demonstrates that while *TECR* is ubiquitously expressed, *TECRL* is predominantly expressed in the heart and skeletal muscle (Fig 2I). Immunofluorescence analysis demonstrated that in COS-1 cells, MYC epitope-tagged mouse Tecrl protein primarily resides in the ER (Fig 2J), which was also confirmed in H10 cells by co-localization with calnexin, an ER chaperone (Fig 2K–M).

Collectively, these results demonstrated that *Tecrl* encodes an ER protein expressed preferentially in the heart and that it is evolutionarily conserved, suggesting an important role in the heart.

### Derivation of patient-specific hiPSCs and differentiation to CMs

Skin biopsies were obtained from patient IV:13 carrying the homozygous *TECRL*c.331+1G>A mutation, his heterozygous father, III:3, and a family member, IV:7 who does not carry the mutation.

Dermal fibroblasts (Fig 3A; left image) were reprogrammed to hiPSCs with Sendai virus vectors encoding the four transcription factors OCT4, SOX2, KLF4, and MYC. Colonies obtained after reprogramming (Fig 3A; center image) morphologically resembled hESC colonies. hiPSC clones expressed the pluripotency markers, NANOG and SSEA4 (Fig 3A; right image; Appendix Fig S2), and had a normal karyotype (Fig 3B; left image; Appendix Fig S2). One line per genotype was selected for further experiments. The transcriptional profile of the hiPSC lines used in the study was analyzed by PluriTest (Müller *et al*, 2011), which showed a high pluripotency score and low novelty score as expected (Fig 3B; right image). The pluripotent nature of the hiPSCs was further confirmed by their ability to differentiate to derivatives of all three germ layers (Fig 3C). Importantly, nucleotide sequence analysis confirmed the presence of a homozygous c.331+1G>A mutation in TECRL$_{Hom}$-hiPSCs (Fig 3D).

### A mutation at the exon 3-intron 3 boundary of *TECRL* leads to exon 3 skipping

hiPSCs were differentiated to the cardiac lineage with a monolayer protocol (Fig 3E), and all three lines produced similar percentages of CMs. Contractile areas were observed in culture at day 10 of differentiation. The structure and organization of sarcomeres in TECRL$_{Hom}$-hiPSC-CMs, TECRL$_{Het}$-hiPSC-CMs and CTRL-hiPSC-CMs was similar based on ACTN2 immunostaining (Fig 3F).

To determine the effect of the c.331+1G>A mutation in *TECRL*, cDNA was prepared from hiPSC-CMs and PCR analysis was performed with primers designed to target *TECRL* exons 2–4. A 171-bp product is expected if the mRNA of *TECRL* contains exon 3, whereas a 126-bp product should be present when exon 3 is missing. Analysis of the amplicons from TECRL$_{Het}$-hiPSC-CMs clearly showed two transcripts, a longer product containing exon 3 and a shorter DNA fragment lacking exon 3 (Fig 3G; left image). As expected, CTRL-hiPSC-CMs exclusively yielded the longer PCR product corresponding to 171 bp. On the contrary, PCR analysis of TECRL$_{Hom}$-hiPSC-CMs revealed a single product of 126 bp (Fig 3G; left image), indicating that the c.331+1G>A mutation in the splice donor site of *TECRL* intron 3 causes complete skipping of exon 3. Amplification of the entire *TECRL* coding region showed a shorter PCR product (1,046 bp) of *TECRL* from TECRL$_{Hom}$-hiPSC-CMs compared to the 1,091-bp amplicon of CTRL-hiPSC-CMs (Fig 3G;

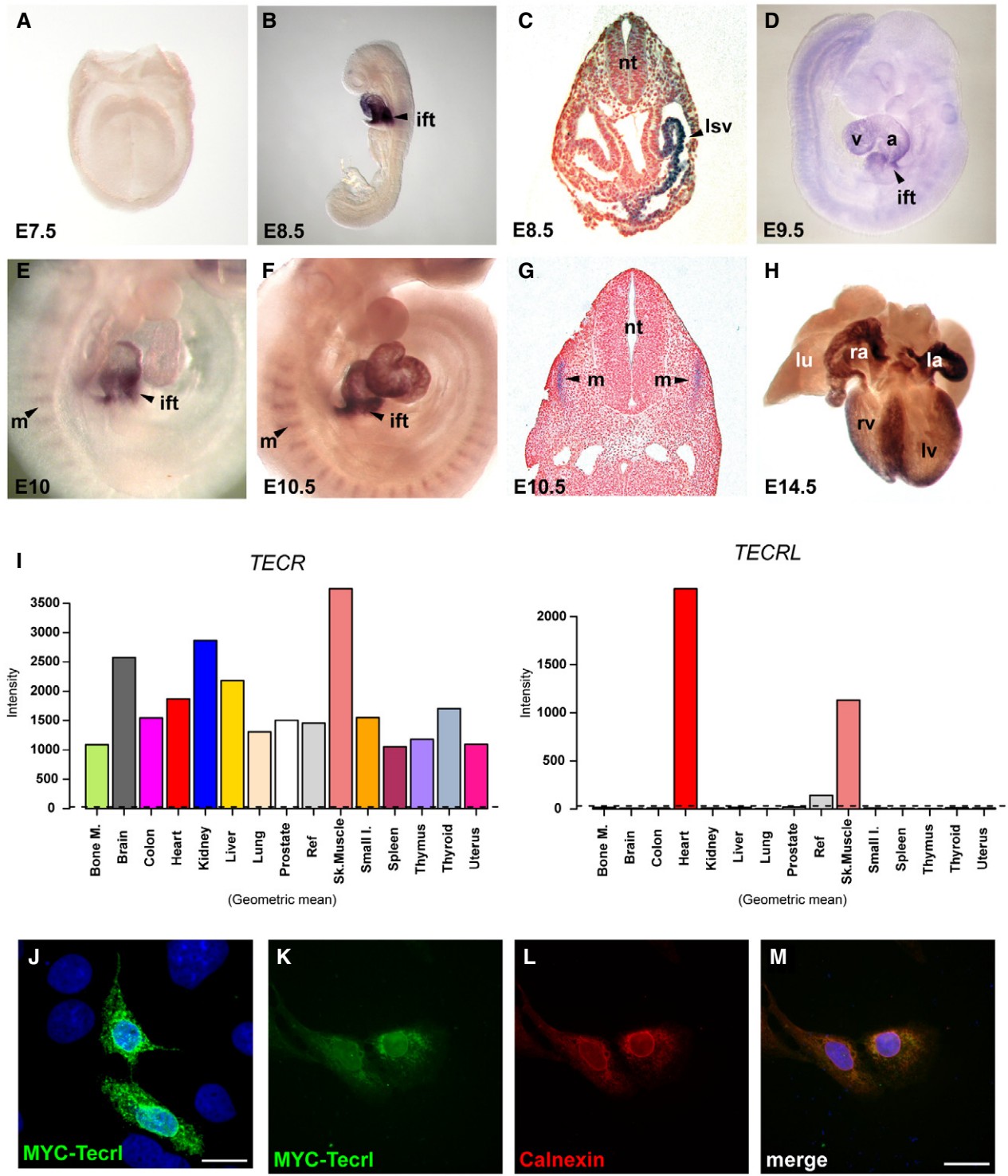

**Figure 2. Spatiotemporal, tissue, and sub-cellular expression analysis of *TECRL*.**

A–H  Expression of *Tecrl* in mouse development. Expression is not observed in (A) the cardiac crescent at E7.5 but is prominent in (B) the inflow tract (ift) region at E8.5, particularly in (C) the left *sinus venosus* (lsv). (D) At E9.5, *Tecrl* expression is observed in all four chambers of the heart but is strongest in the inflow tract. (E–G) From E10 onwards, *Tecrl* is expressed in the somites. (H) At E14.5, *Tecrl* expression is present in the entire myocardium. Atrium (a), ventricle (v), myotome (m), neural tube (nt), lung (lu), right atrium (ra), left atrium (la), right ventricle (rv), and left ventricle (lv).

I  mRNA expression analyses of *TECR* and *TECRL* by RT–qPCR in human tissues demonstrate preferential expression of *Tecrl* in the heart. Bone marrow (Bone M.), heart, skeletal muscle (Sk.Muscle), uterus, liver, spleen, thymus, thyroid, prostate, brain, lung, small intestine (Small I.), and colon.

J  Localization of MYC-Tecrl in COS-1 cells. The accumulation of MYC-Tecrl is perinuclear, consistent with it being localized to the ER. Scale bar: 10 μm.

K–M  Co-localization of MYC-Tecrl and the ER marker calnexin in H10 cells. Nuclei stained with DAPI. Scale bar: 10 μm.

**Figure 3.   Generation of hiPSCs and differentiation to cardiomyocytes.**

A   Skin fibroblasts (left) from IV:13, homozygous for the *TECRL*c.331+1G>A mutation, were reprogrammed to hiPSCs (center), which expressed the pluripotency markers NANOG and SSEA4 (right). Scale bars: 100 μm.

B   TECRL$_{Hom}$-hiPSCs have a normal karyotype (left) and PluriTest demonstrates a high pluripotency score and low novelty score for all three hiPSC lines (right).

C   TECRL$_{Hom}$-hiPSCs generate derivatives of mesoderm (left), endoderm (center), and ectoderm (right). Scale bars: 25 μm.

D   DNA sequencing confirms *TECRL*c.331+1G>A in the affected family members. Control (left); heterozygous (center); homozygous (right).

E   Schematic of the protocol used for cardiac differentiation of hiPSCs.

F   ACTN2 immunostaining of CTRL-, TECRL$_{Het}$-, and TECRL$_{Hom}$-hiPSC-CMs. Scale bar: 25 μm.

G   Analysis of RT–PCR products by gel electrophoresis. Amplification products of *TECRL* exons 2–4 from CTRL-, TECRL$_{Het}$-, and TECRL$_{Hom}$-hiPSC-CMs (left) and coding sequence of *TECRL* from CTRL- and TECRL$_{Hom}$-hiPSC-CMs (right).

Source data are available online for this figure.

right image). Nucleotide sequencing analysis of the PCR product from TECRL$_{Hom}$-hiPSC-CMs confirmed deletion of 45 bp corresponding to exon 3.

## TECRL$_{Hom}$-hiPSC-CMs exhibit abnormalities in calcium handling

Impaired calcium homeostasis underlies the pathophysiology of various IADs, particularly CPVT. To investigate whether the *TECRL*c.331+1G>A mutation has an effect on canonical calcium-handling proteins, we investigated their expression at the protein level by Western blotting. There was a 52% reduction in RYR2 protein and 85% reduction in CASQ2 protein in TECRL$_{Hom}$-hiPSC-CMs, while SERCA2a, PLB, NCX, and Ca$_v$1.2 protein levels were unaffected (Appendix Fig S3).

Next, we studied intracellular calcium ([Ca$^{2+}$]$_i$) transients of TECRL$_{Hom}$-hiPSC-CMs. Fig 4A shows typical [Ca$^{2+}$]$_i$ transient recordings of CTRL-hiPSC-CMs, TECRL$_{Het}$-hiPSC-CMs, and TECRL$_{Hom}$-hiPSC-CMs following 1-Hz electrical stimulation. A summary of average [Ca$^{2+}$]$_i$ characteristics is shown in Fig 4B and C. The [Ca$^{2+}$]$_i$ transient rose markedly slower in TECRL$_{Hom}$-hiPSC-CMs than in CTRL-hiPSC-CMs (Fig 4A) resulting in a significant difference in the time required to reach 50% of the [Ca$^{2+}$]$_i$ transient amplitude, t$_{1/2}$ (Fig 4B). While there were no significant differences in systolic [Ca$^{2+}$]$_i$, the diastolic [Ca$^{2+}$]$_i$ was markedly higher in TECRL$_{Hom}$-hiPSC-CMs compared with CTRL-hiPSC-CMs (Fig 4B). Consequently, the amplitude of the [Ca$^{2+}$]$_i$ transient was lower in TECRL$_{Hom}$-hiPSC-CMs (Fig 4C). Furthermore, the decay of the [Ca$^{2+}$]$_i$ transient was remarkably slower in TECRL$_{Hom}$-hiPSC-CMs, as indicated by the significant increase in the time constant (tau) of decay (Fig 4C).

To gain further insight into the observed differences in [Ca$^{2+}$]$_i$ transient properties of TECRL$_{Hom}$-hiPSC-CMs and ascertain the cause of slower decay time, we measured the activity of sarcoplasmic reticulum (SR) Ca$^{2+}$ ATPase (SERCA), Na$^+$-Ca$^{2+}$ exchanger (NCX), and the slow mechanisms (mitochondrial Ca$^{2+}$ uniporter and sarcolemmal Ca$^{2+}$ ATPase) in Ca$^{2+}$ extrusion from the cytoplasm by application of the RYR2 agonist, caffeine, and the NCX1 blocker, NiCl$_2$ (Bassani & Bers, 1995; Díaz *et al*, 2004). About 10 mM caffeine induced fast release of Ca$^{2+}$ from the SR into the cytosol and SERCA activity ($K_{SERCA}$) was analyzed by comparing the rates of decay of spontaneous [Ca$^{2+}$]$_i$ transients ($K_{sys}$) and caffeine-evoked transients ($K_{caff}$). The role of NCX in Ca$^{2+}$ removal from the cytoplasm was obtained through subtraction of the rate of decay of the caffeine-evoked [Ca$^{2+}$]$_i$ transients ($K_{caff}$) from that of caffeine-evoked [Ca$^{2+}$]$_i$ transients in the presence of 10 mM NiCl$_2$

($K_{Caff+Ni}$). Representative [Ca$^{2+}$]$_i$ transients of a CTRL-hiPSC-CM and TECRL$_{Hom}$-hiPSC-CM are shown in Fig 5A.

The amplitude of caffeine-evoked [Ca$^{2+}$]$_i$ transients in the presence of NiCl$_2$ was considerably lower in TECRL$_{Hom}$-hiPSC-CMs compared with CTRL-hiPSC-CMs (Fig 5B). However, fractional Ca$^{2+}$ release, which is a measure of the amplitude of the normal systolic [Ca$^{2+}$]$_i$ transient as a fraction of the amplitude of caffeine-evoked [Ca$^{2+}$]$_i$ transient in the presence of NiCl$_2$, did not differ significantly between the groups (Fig 5B). $K_{SERCA}$ and $K_{NCX}$ were significantly lower in TECRL$_{Hom}$-hiPSC-CMs compared with CTRL-hiPSC-CMs (Fig 5C). $K_{slow\ mechanisms}$ indicated by the rate of Ca$^{2+}$ transient decay in the presence of caffeine, and NiCl$_2$ ($K_{Caff+Ni}$) was not affected in TECRL$_{Hom}$-hiPSC-CMs (Fig 5C).

Although SERCA and NCX activities were lower in TECRL$_{Hom}$-hiPSC-CMs, the relative contribution of SERCA, NCX, and slow mechanisms to [Ca$^{2+}$]$_i$ transient decay did not differ significantly between the groups (Fig 5D). SERCA and NCX removed ≈65–70% and 25–30% of the activator Ca$^{2+}$ from the cytosol, respectively, whereas the mitochondrial Ca$^{2+}$ uptake and sarcolemmal Ca$^{2+}$-ATPase accounted for the removal of < 8% of [Ca$^{2+}$]$_i$.

The [Ca$^{2+}$]$_i$ transient properties of TECRL$_{Het}$-hiPSC-CMs were largely in between CTRL-hiPSC-CMs and TECRL$_{Hom}$-hiPSC-CMs, suggesting a gene dosage dependency (Figs 4B and C, and 5B–D).

## TECRL$_{Hom}$-hiPSC-CMs and hESC-CMs with *TECRL* knockdown exhibit prolonged APs

Increased diastolic [Ca$^{2+}$]$_i$ is believed to be a substrate for delayed afterdepolarizations (DADs) leading to triggered arrhythmias which maybe further aggravated by catecholaminergic stimulation. TECRL$_{Hom}$-hiPSC-CMs showed elevated diastolic [Ca$^{2+}$]$_i$ which prompted us to investigate their action potential (AP) properties. We also studied the AP parameters of hESC-CMs with *TECRL* knockdown in order to compare their electrophysiological phenotype to TECRL$_{Hom}$-hiPSC-CMs.

First, we studied the AP properties of TECRL$_{Hom}$-hiPSC-CMs (Fig 6A). Representative APs and averaged AP parameters of hiPSC-CMs from CTRL, TECRL$_{Het}$, and TECRL$_{Hom}$ lines paced at 1-Hz are shown in Fig 6B and C. Average resting membrane potential (RMP), maximum upstroke velocity (d$V$/d$t_{max}$), maximal AP amplitude (APA$_{max}$), and AP plateau amplitude (APA$_{plat}$) did not differ significantly between the three groups (Fig 6C). However, APs of TECRL$_{Hom}$-hiPSC-CMs displayed marked prolongation of AP duration (APD) at 20% repolarization (APD$_{20}$) and there was a clear

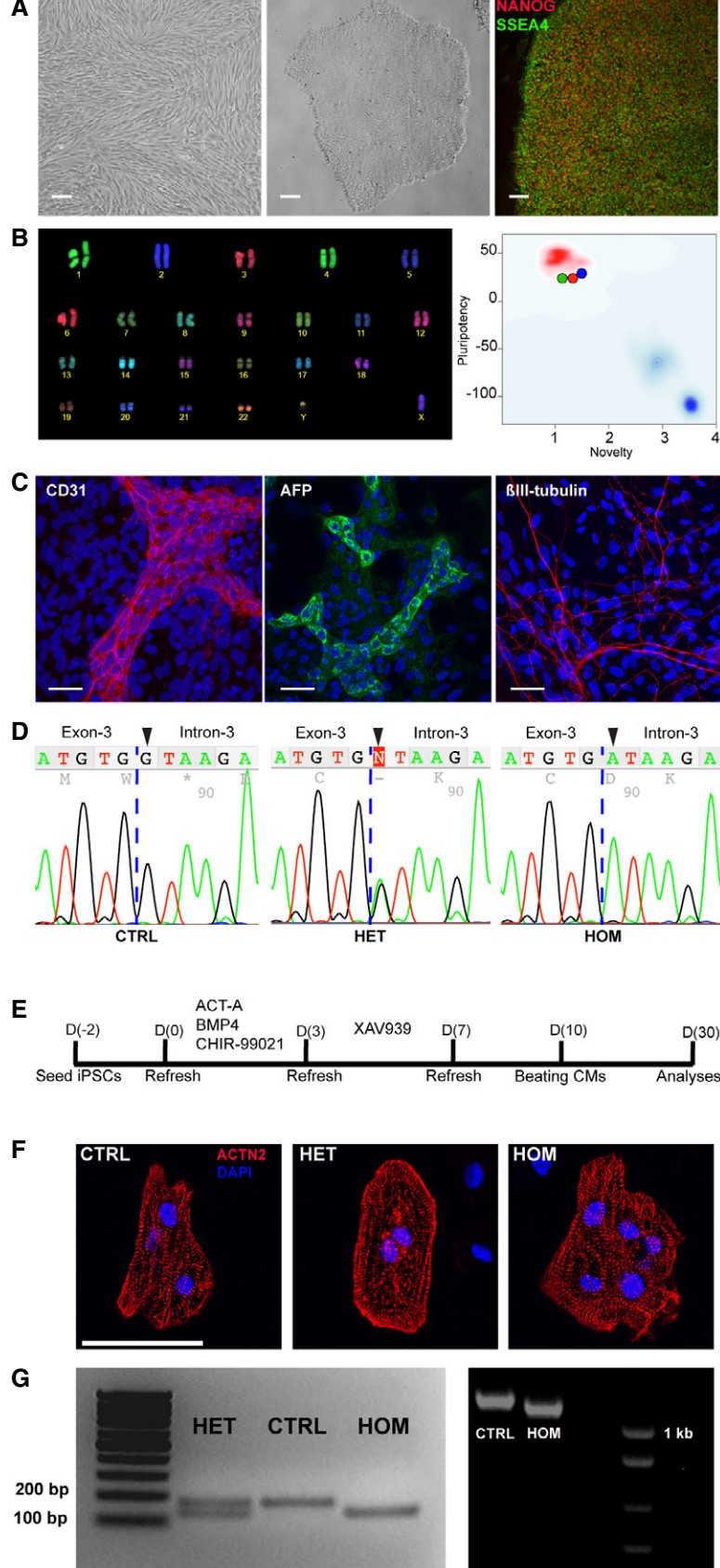

**Figure 3.**

trend for an increase in APD at 50% ($APD_{50}$) and 90% ($APD_{90}$) repolarization compared to CTRL-hiPSC-CMs (Fig 6C).

Next, we performed lentiviral short hairpin (sh) RNA-mediated knockdown of *TECRL* in hESC-CMs to study their AP properties. Lentiviral vectors encoding five different *TECRL*-specific shRNAs (Appendix Table S3) for *TECRL* were tested in hESC-CMs, two of which (*TECRL*-sh#3 and *TECRL*-sh#4) gave efficient knockdown as assessed by RT–qPCR. Primer sequences are provided in Appendix Table S4. However, following puromycin selection, hESC-CMs

treated with *TECRL*-sh#4 showed pronounced cytotoxicity (Appendix Fig S4A). Therefore, *TECRL*-sh#3 (hereinafter referred to as sh*TECRL*) was selected for further experiments and transduction in hESC-CMs resulted in 70% reduction in *TECRL* mRNA level in comparison with cells exposed to a control vector (sh*CONTR*; Appendix Fig S4B). hESC-CMs transduced with sh*CONTR* or sh*TECRL* maintained their cardiac phenotype and knockdown of *TECRL* in hESC-CMs did not affect *TNNT2* expression (Appendix Fig S4B). AP parameters of sh*CONTR*- or sh*TECRL*-

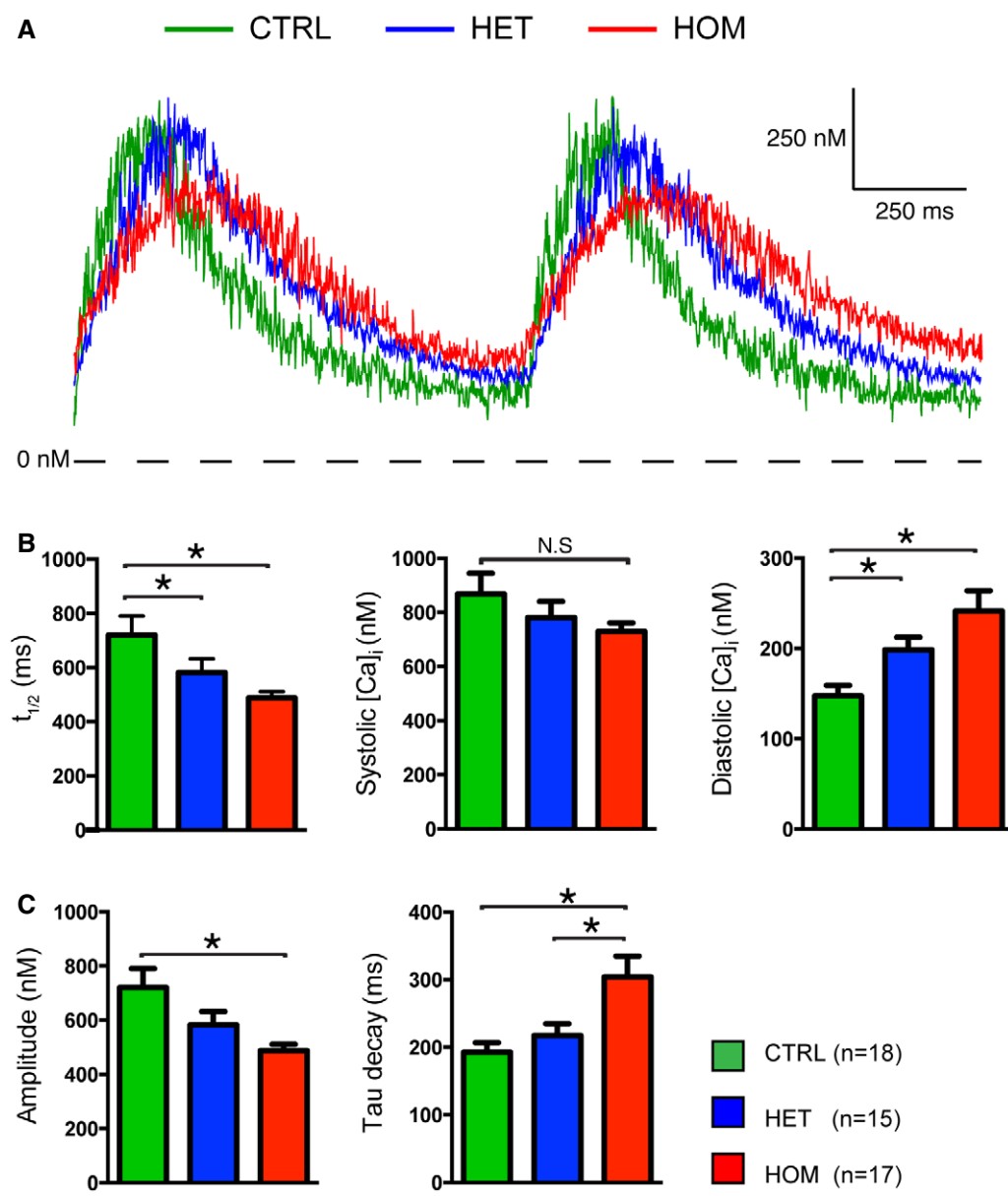

**Figure 4. Whole-cell $[Ca^{2+}]_i$ transients in hiPSC-CMs.**

A   Representative traces of $[Ca^{2+}]_i$ transients in Indo-1 AM-loaded hiPSC-CMs paced at 1 Hz.
B   Time to reach 50% of $[Ca^{2+}]_i$ transient amplitude ($t_{1/2}$) and $[Ca^{2+}]_i$ concentrations during systole and diastole.
C   Amplitude and tau decay of the $[Ca^{2+}]_i$ transient.

Data information: $n$ = 5–6 cells each from three independent experiments (mean ± SEM); *$P < 0.05$, statistical significance was assessed with one-way ANOVA on ranks followed by Dunn's test in case of failed normality or variance. Otherwise, one-way ANOVA followed by Student–Newman–Keuls test was used.

    

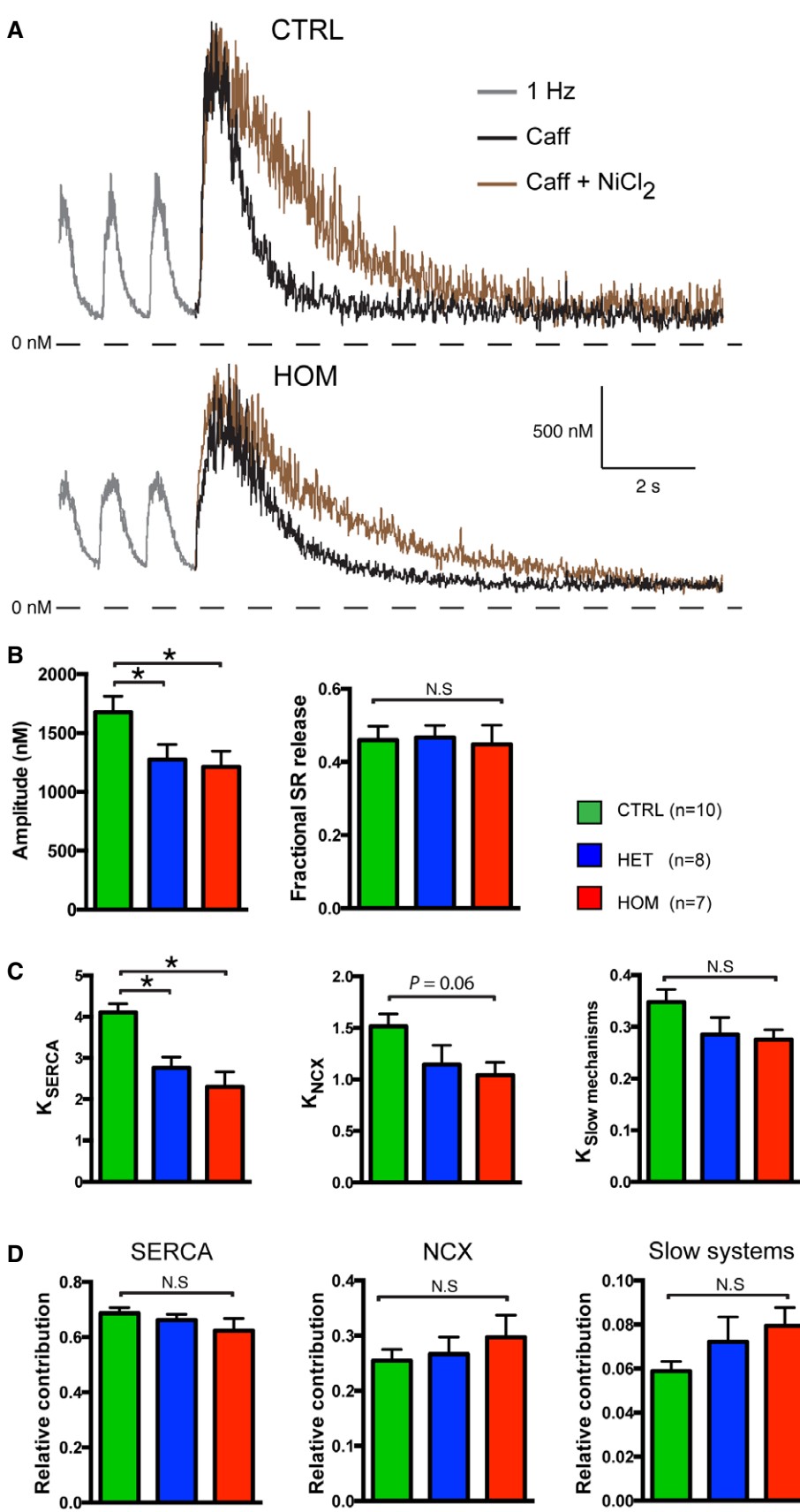

**Figure 5.**

◄

**Figure 5.    $[Ca^{2+}]_i$ extrusion mechanisms in hiPSC-CMs.**

A    Representative traces of $[Ca^{2+}]_i$ transients in Indo-1 AM-loaded hiPSC-CMs paced at 1 Hz, in the presence of caffeine (caff) or caffeine and $NiCl_2$.
B    Amplitude of $[Ca^{2+}]_i$ in the presence of caffeine and $NiCl_2$ and fractional SR release in hiPSC-CMs.
C    Rate constants of SERCA-, NCX-, and slow mechanism-based $[Ca^{2+}]_i$ decay in hiPSC-CMs.
D    Relative contribution of SERCA, NCX, and slow mechanisms to $[Ca^{2+}]_i$ extrusion in hiPSC-CMs.

Data information: $n$ = 2–3 cells each from three independent experiments (mean ± SEM); *$P < 0.05$, statistical significance was assessed with one-way ANOVA on ranks followed by Dunn's test in case of failed normality or variance. Otherwise, one-way ANOVA followed by Student–Newman–Keuls test was used.

transduced hESC-CMs stimulated at 1-Hz were recorded, and representative traces as well as averaged data are shown in Appendix Fig S4C and D. RMP, $dV/dt_{max}$, $APA_{max}$, and $APA_{plat}$ did not differ significantly between the two experimental groups (Appendix Fig S4D). However, hESC-CMs treated with sh*TECRL* displayed significantly prolonged APs as evidenced by the increase in $APD_{20}$, $APD_{50}$, and $APD_{90}$ values (Appendix Fig S4D). In addition, Western blotting for canonical calcium-handling proteins revealed a 56% decrease in RYR2 protein and 18% decrease in CASQ2 in sh*TECRL*-treated hESC-CMs while SERCA2a, PLB, NCX1, and $Ca_v1.2$ protein levels were unaffected (Appendix Fig S4E).

### TECRL$_{Hom}$-hiPSC-CMs have an increased susceptibility to triggered activity

To evaluate the susceptibility of TECRL$_{Hom}$-hiPSC-CMs to triggered activity, we applied a fast pacing episode (3 Hz; 10 s), followed by a 10-s pause in the absence or presence of 10 nM noradrenaline (NA). Representative traces of CTRL-hiPSC-CMs and TECRL$_{Hom}$-hiPSC-CMs, in the absence and presence of NA, are shown in Fig 7A and C, respectively. In the absence of NA, the last stimulated AP (arrow) is followed by APs which seem to be due to diastolic depolarization rather than DADs (Fig 7A) since the frequency of such "triggered" APs did not differ significantly between the experimental groups (Fig 7B). Moreover, the prevalence of "triggered" and spontaneous APs (i.e., APs elicited without a fast pacing protocol) was also found to be similar in the absence of NA (Fig 7B), which further suggests that the "triggered" APs in Fig 7A are not due to DADs. However, in the presence of NA, "triggered" APs were abundantly present in TECRL$_{Hom}$-hiPSC-CMs compared with CTRL-hiPSC-CMs (Fig 7C). The incidence of "triggered" APs was also significantly higher than that of spontaneous APs (Fig 7D), indicating that the higher frequency of "triggered" APs in TECRL$_{Hom}$-hiPSC-CMs is due to DADs rather than spontaneous activity. Similar to what was observed for $[Ca^{2+}]_i$ transient properties, AP properties and susceptibility to triggered activity of TECRL$_{Het}$-hiPSC-CMs were largely in between those of CTRL-hiPSC-CMs and TECRL$_{Hom}$-hiPSC-CMs.

### Effect of flecainide on hiPSC-CMs

Flecainide, a class Ic antiarrhythmic drug, has been shown to be effective in CPVT patients, (Watanabe *et al*, 2009; van der Werf *et al*, 2011) although its precise mechanism of action has been a point of debate. To investigate whether flecainide has an effect on triggered activity of TECRL$_{Hom}$-hiPSC-CMs, we administered 5 μM of the drug and analyzed AP properties.

Figure 8A shows typical APs of a TECRL$_{Hom}$-hiPSC-CM in the absence and presence of flecainide. The average effects of flecainide

on AP parameters in all three hiPSC-CM groups are summarized in Fig 8B. Flecainide reduced the $dV/dt_{max}$ and caused AP prolongation in all three groups without affecting the RMP (Fig 8B). These effects did not differ significantly between the groups. Figure 8C shows typical examples of triggered (top) and spontaneous (bottom) APs of a TECRL$_{Hom}$-hiPSC-CM in the absence and presence of flecainide. The averaged effects of the drug on triggered and spontaneous activity in the three experimental groups are summarized in Fig 8D. While flecainide reduced the frequency of both triggered and spontaneous activity (Fig 8D), these reductions were not significantly different between the three groups. However, the reduction in triggered activity was more pronounced than that of spontaneous activity in TECRL$_{Het}$-hiPSC-CMs and TECRL$_{Hom}$-hiPSC-CMs (Fig 8D). These results suggest that a treatment regimen with flecainide might be effective in preventing arrhythmias in patients carrying the *TECRL*c.331+1G>A mutation.

## Discussion

The etiology of a number of IADs remains unknown, and similar diagnostic challenges are seen in other cardiac diseases. For example, LQTS is a condition with monogenic inheritance for which hundreds of rare mutations have been reported in 16 genes (LQTS1-16). LQTS patients are grouped into distinct syndromes based on their clinical characteristics as well as their genotypes. However, 30% of patients are without a genetically confirmed diagnosis and remain at risk of cardiac events and sudden death (Tester & Ackerman, 2006). Similarly, in CPVT, 35–45% of the patients without a known genetic cause remain at risk for adverse arrhythmic events (Ackerman *et al*, 2011). WES has emerged as a means to meet this challenge (Biesecker & Green, 2014). In the current study, we used WES to identify *TECRL* as a novel life-threatening inherited arrhythmia gene associated with a recessive form of inherited arrhythmia with a clinical phenotype that has overlapping features of LQTS and CPVT.

Clinically, affected individuals from the Sudanese family and both French Canadian cases presented with stress-induced ventricular arrhythmias including SCD or aborted SCD. At rest, the QTc was normal or mildly prolonged while pharmacological sympathetic stimulation resulted in paradoxical QT prolongation as demonstrated in the French Canadian cases. Notably, affected individuals in the Sudanese family presented arrhythmias in childhood while the onset of symptoms in both French Canadian patients was in early adulthood. The most likely explanation for this difference in the age of onset is the more drastic effect of the splice site mutation identified in the Sudanese family. In all cases, the disease is highly penetrant and life-threatening, with all affected individuals having experienced cardiac arrest at a young age. The French Canadian

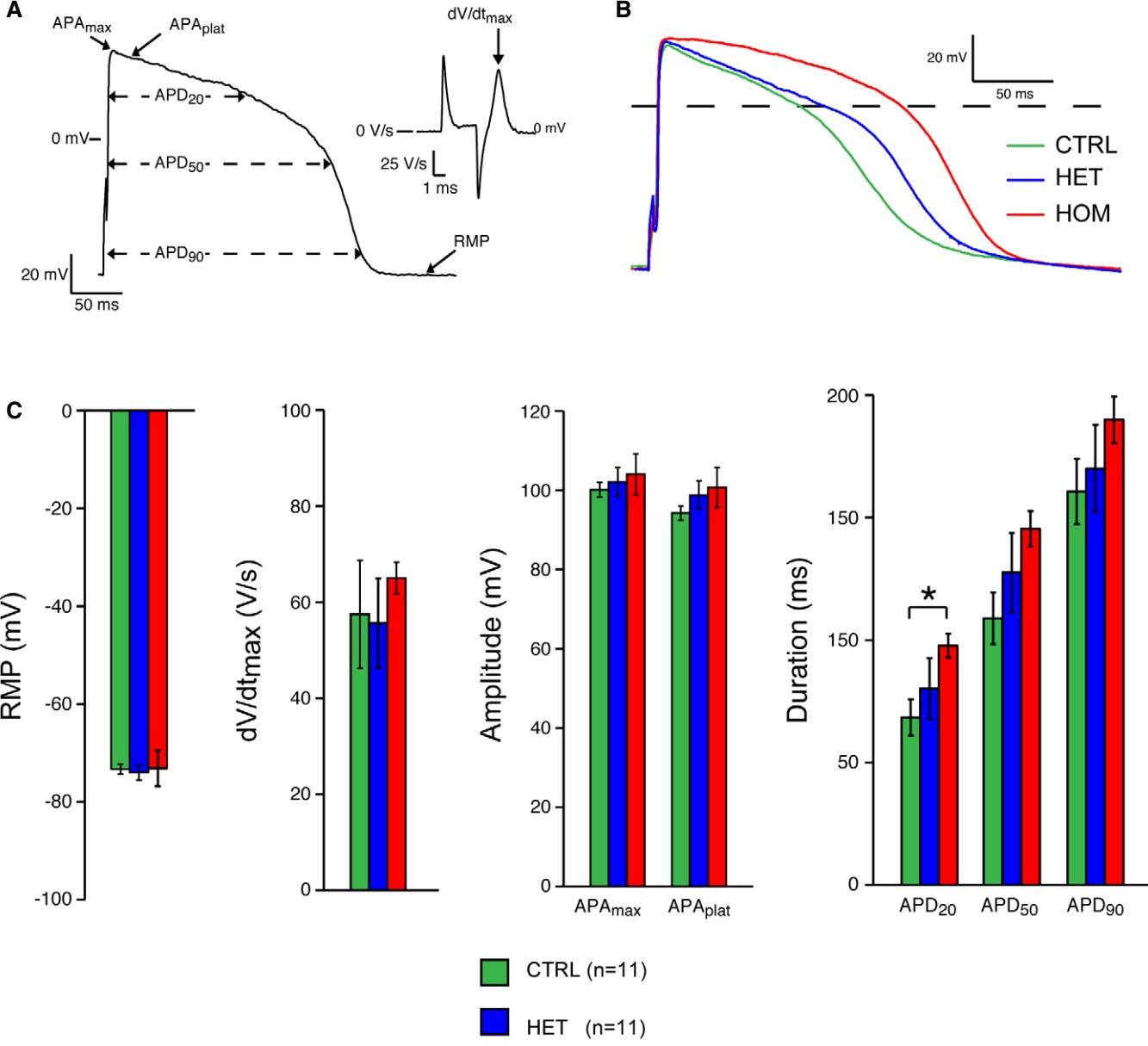

**Figure 6.  AP characteristics of TECRL-hiPSC-CMs.**

A   AP illustrating the analyzed parameters.

B   Representative APs from control (CTRL), heterozygous (HET), and homozygous (HOM) CMs.

C   RMP, dV/dt$_{max}$, APAmax, APAplat, APD$_{20}$, APD$_{50}$, and APD$_{90}$ of CTRL-, TECRL$_{Het}$-, and TECRL$_{Hom}$-hiPSC-CMs.

Data information: $n$ = 3–4 cells each from three independent experiments (mean $\pm$ SEM); *$P$ < 0.05, statistical significance was assessed with one-way ANOVA on ranks followed by Dunn's test in case of failed normality or variance. Otherwise, one-way ANOVA followed by Student–Newman–Keuls test was used. AP = action potential; APA$_{max}$ = maximum AP amplitude, APA$_{plat}$ = AP plateau amplitude; APD$_{20}$, APD$_{50}$, and APD$_{90}$ = AP duration at 20, 50, and 90% repolarization, respectively; dV/dt$_{max}$ = maximum upstroke velocity; RMP = resting membrane potential.

cases also showed recurrent stress-induced AT, which sometimes triggered VT. In contrast, no atrial arrhythmia was documented in the Sudanese family. This may be due to the shorter follow-up related to the high lethality of the disease, as well as differences in monitoring (the French Canadian patients have a dual-chamber ICDs, with continuous monitoring of the atrial ECG). The heart was structurally normal in all affected individuals, with the exception of two individuals in the Sudanese family that also had tetralogy of Fallot. Whether these congenital heart defects are part of the same syndrome or are caused by another genetic defect in this consanguineous family remains unclear. During longer follow-up of the French Canadian cases, it was noted that less potent beta-blockers

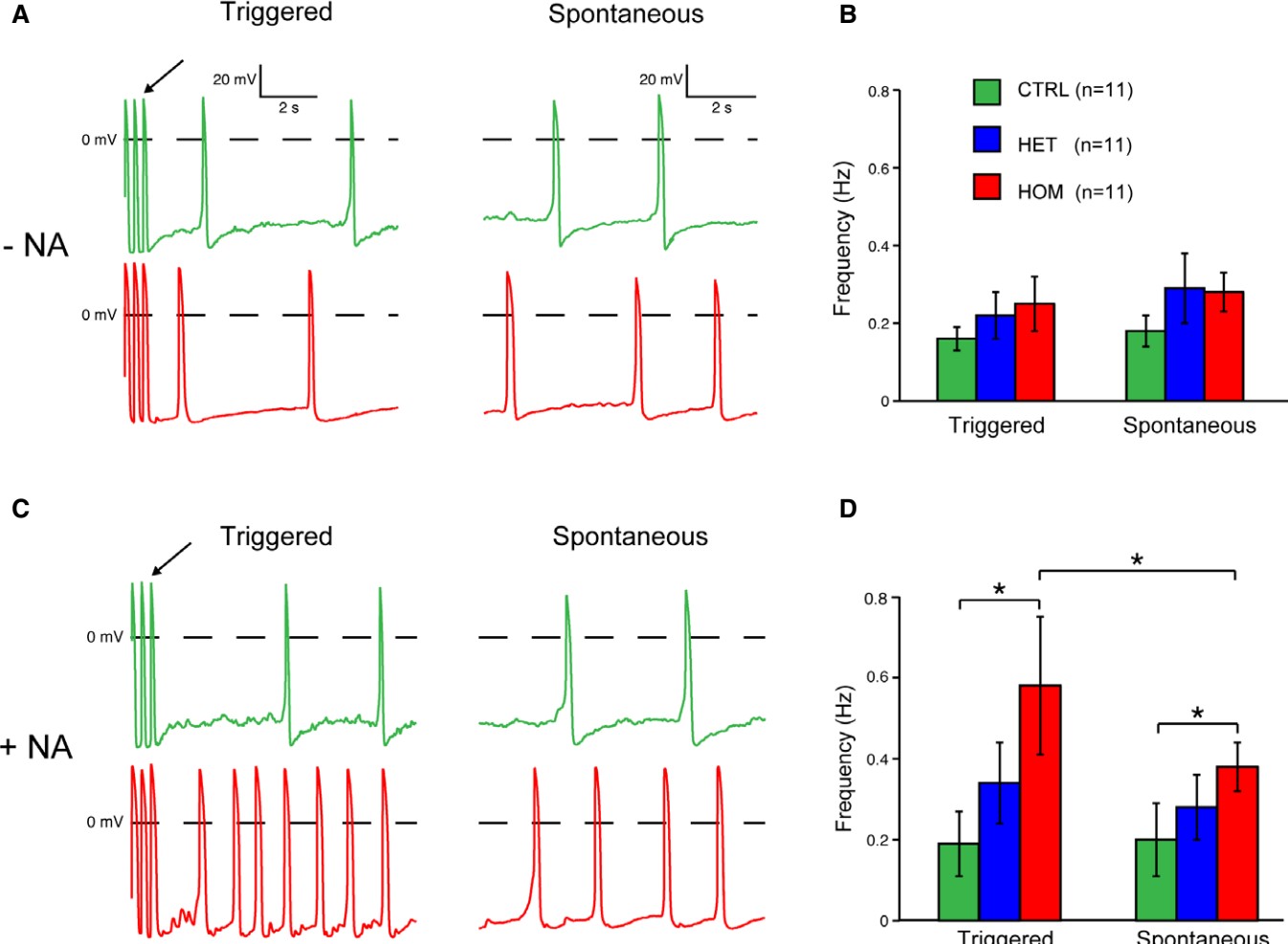

**Figure 7. TECRL_Hom-hiPSC-CMs demonstrate increased susceptibility to triggered activity in response to NA.**

A, B   Representative AP traces (A) and averaged activity (B) of triggered and spontaneous activity in hiPSC-CMs in the absence of NA.
C, D   Representative AP traces (C) and averaged activity (D) of triggered and spontaneous activity in hiPSC-CMs in the presence of NA.

Data information: *n* = 3–4 cells each from three independent experiments (mean ± SEM); *P < 0.05, statistical significance within a group before and after application of NA was assessed with paired *t*-test or Wilcoxon signed rank test in case of failed normality; arrows in (A) and (C) indicate last stimulated AP. NA = noradrenaline.

(metoprolol and bisoprolol) are associated with arrhythmia recurrence, while maximal tolerated doses of nadolol were highly effective.

Whether the clinical phenotype should be classified as LQTS or CPVT is a subject of discussion. Despite having nearly identical phenotypes, the Sudanese family was diagnosed with CPVT, while the French Canadian cases were diagnosed with LQTS. The arrhythmias were induced by catecholaminergic stimulation, and most affected individuals had a normal baseline QTc, which together suggest CPVT. In contrast, mild QTc prolongation in some patients and the paradoxical QT prolongation during catecholaminergic stimulation indicate a LQTS phenotype (Obeyesekere *et al*, 2011). Such an overlapping clinical phenotype is also observed in calmodulin-related inherited arrhythmia (Makita *et al*, 2014), as well as in *KCNJ2* mutations causing Andersen-Tawil syndrome, which can present as LQTS or CPVT phenocopies (Kimura *et al*, 2012). Therefore, we recommend that a

homozygous mutation in *TECRL* should be considered as a possible cause of the etiology in patients presenting with stress-induced complex ventricular arrhythmias or cardiac arrest at a young age, whether they are diagnosed with (gene-elusive) LQTS or CPVT. The identification of additional families with the same syndrome and genetic defect could help in better characterizing/categorizing the phenotype.

Using murine and human tissues as well as cell lines, we showed that *TECRL* expression is restricted to cardiac and skeletal muscles and that most of the TECRL protein is localized to the ER, which is a critical site of protein, lipid, and glucose metabolism as well as calcium homeostasis. Accordingly, reactome pathway analysis indicates a role for *TECRL* in fatty acid and lipid metabolism while gene ontology annotates its function as an enzyme involved in catalysis of redox reactions. In the adult heart, free fatty acids are the primary substrate for energy production at rest and abnormalities in fatty acid oxidation have been linked to arrhythmias (Bonnet *et al*, 1999).

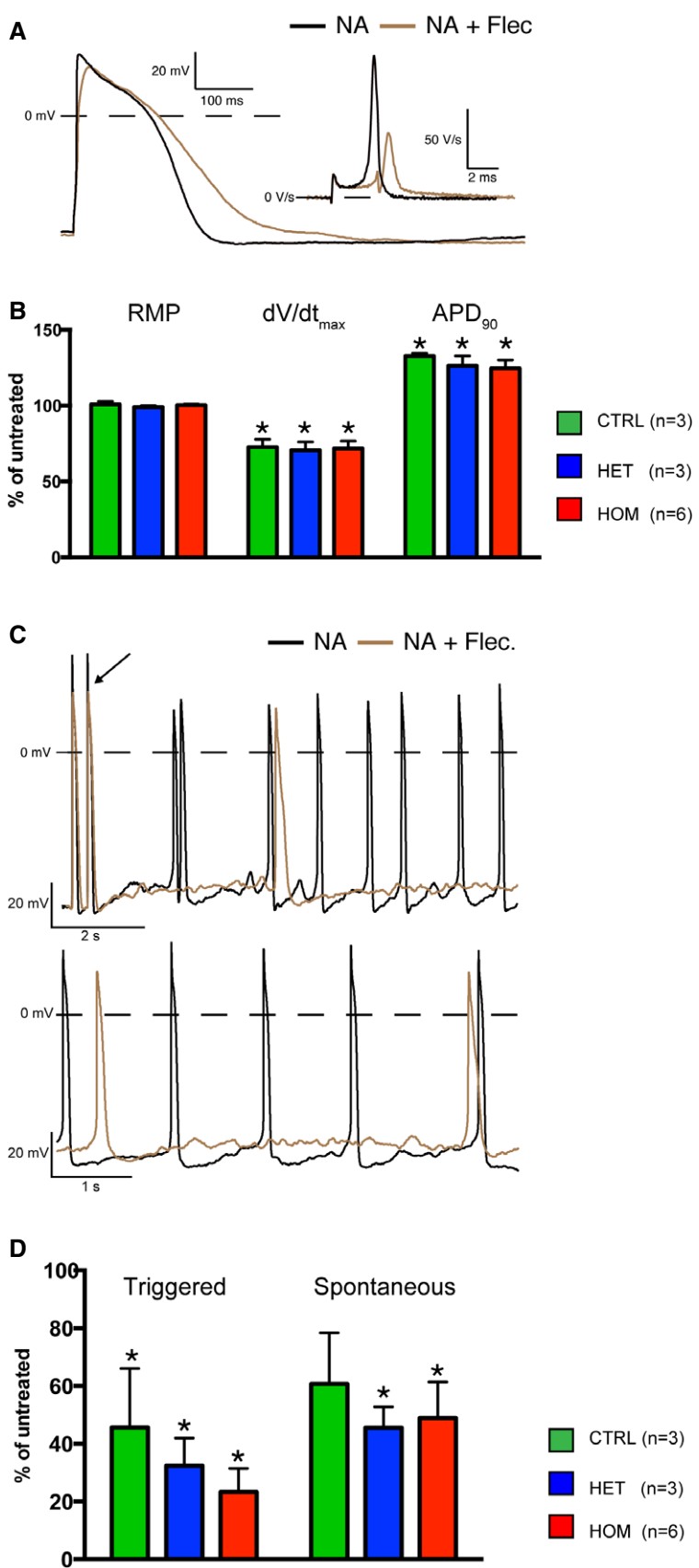

Figure 8.

◀

**Figure 8. Flecainide alleviates triggered activity in TECRL$_{Hom}$-hiPSC-CMs.**

A    Representative AP trace of a TECRL$_{Hom}$-hiPSC-CM in the presence of NA alone or NA and flecainide.

B    Effect of flecainide on AP parameters of hiPSC-CMs.

C, D    Addition of 5 μM of flecainide decreased the susceptibility to triggered activity in hiPSC-CMs. Note that the effect of flecainide on triggered activity of TECRL$_{Het}$- and TECRL$_{Hom}$-hiPSC-CMs is more pronounced than its effect on their spontaneous activity.

Data information: *n* = 1–2 cells each from three independent experiments (mean ± SEM); *$^*P < 0.05$, statistical significance within a group before and after application of flecainide was assessed with paired *t*-test; arrow in (C) indicates last stimulated AP. NA = noradrenaline, Flec = flecainide.

Perturbations in physiological levels of lipids/fatty acids and metabolic function can have direct consequences on ion channels and calcium-handling proteins (Charnock, 1994; Boland & Drzewiecki, 2008; Barth & Tomaselli, 2009). Further studies are necessary to elucidate the role of *TECRL* in lipid metabolism and how mutations are linked to cardiac arrhythmias.

hiPSCs offer important opportunities to model cardiac diseases *in vitro* and are especially valuable when myocardial tissue of the patient is not accessible to determine the consequences of a particular mutation. Using hiPSC-CMs, we demonstrated that the *TECRL* mRNA in TECRL$_{Hom}$-hiPSC-CMs lacks exon 3 and that these cells recapitulate salient features of the disease phenotype.

Altered calcium homeostasis is a hallmark of CPVT phenotype, and mutations in *RYR2* or *CASQ2* have been identified to cause this disease. Interestingly, protein levels of RYR2 and CASQ2 were significantly downregulated in TECRL$_{Hom}$-hiPSC-CMs. We also found a smaller $[Ca^{2+}]_i$ transient amplitude in these CMs. Since the $Ca^{2+}$ concentration in the SR is a major regulator of the $[Ca^{2+}]_i$ transient amplitude, this finding suggests a lower SR $Ca^{2+}$ content in TECRL$_{Hom}$-hiPSC-CMs. Indeed, caffeine-evoked transients in the presence of 10 mM NiCl$_2$, which are an indicator of SR $Ca^{2+}$ content, were significantly lower in TECRL$_{Hom}$-hiPSC-CMs, which could be associated with downregulation of RYR2.

Additionally, the diastolic $[Ca^{2+}]_i$ concentration was significantly higher in TECRL$_{Hom}$-hiPSC-CMs than in CTRL-hiPSC-CMs, which could explain their increased propensity to DADs upon adrenergic stimulation with NA. The increase in diastolic $[Ca^{2+}]_i$ in TECRL$_{Hom}$-hiPSC-CMs may in part be reasoned by the observed decrease in SERCA and NCX activity. Although we did not observe a decrease in protein levels of SERCA2A and NCX in TECRL$_{Hom}$-hiPSC-CMs compared with CTRL-hiPSC-CMs, we noted a decrease in their activity by functional studies. It has been established that a number of post-translational modifications regulate the activity of ionophoric proteins, which could clarify the observed effects (Ruknudin *et al*, 2007; Stammers *et al*, 2015). Decreased NCX activity has been shown to reduce DAD amplitude resulting in fewer spontaneous APs (Bögeholz *et al*, 2015). However, in our model, despite a lower NCX activity in TECRL$_{Hom}$-hiPSC-CMs compared with CTRL-hiPSC-CMs, the increased diastolic $[Ca^{2+}]_i$ was sufficient to induce DADs and subsequent triggered APs.

We also assessed the relative contribution of SERCA, NCX, and the slow mechanisms to $[Ca^{2+}]_i$ removal from the cytosol. In isolated adult ventricular CMs, the decline of the $[Ca^{2+}]_i$ transient is mainly due to reuptake of $Ca^{2+}$ into the SR by SERCA and extrusion of $Ca^{2+}$ via sarcolemmal NCX with a minor contribution of the slow mechanisms (Bers, 2006; Dibb *et al*, 2007). Our results correlate with this existing data suggesting that the $Ca^{2+}$ transport systems in hiPSC-CMs are similar to those reported in adult human and animal CMs (Bers, 2000).

At the single cell level, APs of TECRL$_{Hom}$-hiPSC-CMs showed an increase in APD$_{20}$. Although not significant, APD$_{50}$ and APD$_{90}$ were also prolonged, compatible with the borderline QTc prolongation in some of the Sudanese patients or with repolarization abnormalities with QT prolongation occasionally seen on the resting ECG of patient 1. Interestingly, hESC-CMs with *TECRL* knockdown exhibited markedly prolonged APD$_{20}$, APD$_{50}$, and APD$_{90}$. Collectively, these findings are consistent with an overlapping LQTS/CPVT phenotype in TECRL$_{Hom}$-hiPSC-CMs manifested by APD prolongation (LQTS) as well as a disturbed calcium handling and increased propensity for DADs during catecholaminergic stimulation (CPVT).

Finally, we demonstrated that the antiarrhythmic drug flecainide suppressed the incidence of triggered APs in TECRL$_{Hom}$-hiPSC-CMs. Flecainide was first classified as a blocker of Na$^+$ current (I$_{Na}$), and more recently, it has been shown to reduce exercise-induced arrhythmias in CPVT patients (Watanabe *et al*, 2009; van der Werf *et al*, 2011). In agreement with flecainide's effect as I$_{Na}$ blocker, we observed a decrease in AP upstroke velocity. We also observed AP prolongation, as previously reported in isolated human cardiac tissue (Wang *et al*, 1990). The precise mechanisms by which flecainide exerts an antiarrhythmic effect is unclear, but has been proposed to be due to reduced opening of RYR2 channels (Watanabe *et al*, 2009) or I$_{Na}$ blockade resulting in decreased excitability (Liu *et al*, 2011; Bannister *et al*, 2015), or a combination of both (Watanabe *et al*, 2009). Although not conclusive, our results suggest that depression of the upstroke velocity of the AP and the consequential decrease in excitability contributes to the observed reduction in triggered activity. It is encouraging that flecainide reduced the incidence of triggered activity in TECRL$_{Hom}$-hiPSC-CMs. However, some DADs were still observed emphasizing the need for additional or more effective drugs to prevent their occurrence in TECRL$_{Hom}$-hiPSC-CMs. Moreover, the I$_{Kr}$ blocking and consequential QT-prolonging effect of flecainide may potentially offset its beneficial effect on the triggered activity. CMs generated from patient-specific hiPSCs were valuable to study the functional consequences of the c.331+1G>A mutation in *TECRL*. The apparent immature phenotype of hiPSC-CMs (e.g., APD < 200 ms) as opposed to freshly isolated human adult ventricular CMs evidently did not preclude their use in disease modeling.

In conclusion, we identified a novel highly lethal autosomal recessive inherited arrhythmia syndrome caused by homozygous mutations in *TECRL* in patients from three different families. Both clinical data from patients and cellular functional data from hiPSC-CMs point toward overlapping features of LQTS and CPVT with triggered activity-mediated arrhythmogenesis. Further studies are needed to elucidate the exact mechanisms underlying the electrical phenotype and to assess the prevalence of *TECRL* mutations in LQTS, CPVT, and cases of unexplained cardiac arrest. hiPSC lines from more patients or introduction of targeted mutations in healthy

control lines may be of value in this context. Screening for mutations in *TECRL* should be considered in selected cases of suspected autosomal recessive inherited arrhythmia syndromes.

# Materials and Methods

## Ethics statement

The French Canadian subjects described in this study were recruited at the Cardiovascular Genetics Center of the Montreal Heart Institute. Informed consent was obtained for both subjects according to procedures approved by the Montreal Heart Institute Ethics Review Board. Signed informed consent was also obtained from participating patients from the Sudanese family or their guardians, the study adhered to the Declaration of Helsinki, and the research protocol was approved by the Al Ain Medical District Human Research Ethics Committee, College of Medicine, United Arab Emirates (UAE) University. Study on hESCs and hiPSCs was performed in the Netherlands, and their use was approved by the Medical Ethics Committee of Leiden University Medical Center (LUMC).

## Clinical data analysis

### French Canadian subjects
Clinical data, including the resting ECGs and 24-h Holter recordings, were collected and evaluated. Clinical data of family members were evaluated when possible.

### Sudanese family
Clinical data of family members (where possible), including baseline ECGs, exercise ECGs, and 24-h Holter recordings, were collected and evaluated. Family members who experienced sudden cardiac death at young age or who demonstrated arrhythmias were considered affected.

## Exome sequencing

Detailed methods are provided in the Appendix.

### French Canadian subjects
Genomic DNA from both patients was extracted from peripheral blood. A single family member, the father of patient 2, was available for follow-up genetic testing and was found to be heterozygous for the Arg196Gln mutation in *TECRL*.

### Sudanese family
Genomic DNA from individuals III:1, III:2, IV:2, and IV:10 (Fig 1D) was extracted from peripheral blood lymphocytes, and WES was carried out at the Beijing Genomics Institute (BGI; Shenzhen, China).

## Reprogramming of primary fibroblasts to induced pluripotent stem cells

Primary fibroblasts from members of the affected family were isolated from skin biopsies with informed consent under protocols approved by the Al Ain Hospital Ethics Committee for Research, SEHA, Abu Dhabi, UAE. Low-passage skin fibroblasts were reprogrammed to induced pluripotent stem cells using non-integrating Sendai virus vectors encoding OCT4, SOX2, KLF4, and MYC as described previously (Zhang *et al*, 2014). Official identification of the cell lines used in this study is as follows: LUMC0046iTECRL (hiPSCs with heterozygous *TECRL*c.331+1G>A mutation); LUMC0047iCTRL (control hiPSCs); and LUMC0048iTECRL (hiPSCs with homozygous *TECRL*c.331+1G>A mutation).

Of several hiPSC clones with embryonic stem cell morphology those showing robust expression of pluripotency markers, OCT-4, NANOG, SSEA4, and/or TRA-181, as analyzed by flow cytometry or by immunocytochemistry, were selected for further study. These hiPSC lines were karyotyped and subjected to a global assessment of pluripotency by PluriTest (Müller *et al*, 2011).

## Maintenance of hiPSC lines and differentiation to CMs

hiPSC lines were maintained in a feeder-free culture in mTESR1 medium (STEMCELL Technologies) on Matrigel (BD Biosciences) and passaged once a week with 1 mg/ml dispase (Gibco).

To induce cardiac differentiation, cells were seeded in a high density on Matrigel and supplemented with mTESR1 medium. Two days post-seeding, mTESR1 medium was replaced with BPEL medium (Ng *et al*, 2008) containing 20 ng/ml activin-A (R&D systems), 20 ng/ml BMP4 (R&D systems), and 1.5 μmol/l CHIR99021 (Axon Medchem). On day 3, the medium was replaced with BPEL medium containing 5 μmol/l XAV939 (Tocris Biosciences). Cells received BPEL medium on day 7 and every 3–4 days thereafter. Beating CMs were first seen in culture at day 10.

## AP measurements

hiPSC-CMs were dissociated to single cells at day 20 using TrypLE™ Select (Life Technologies) and plated on Matrigel-coated coverslips. APs were recorded 10 days after dissociation with the amphotericin-perforated patch-clamp technique as described previously (Devalla *et al*, 2015) and further detailed in the Appendix.

## Ca²⁺ measurements

hiPSC-CMs were dissociated at day 20, and $[Ca^{2+}]_i$ transients were measured 10 days after dissociation in Indo-1 AM (Molecular Probes, Eugene, OR, USA) loaded cells as described before (van Borren *et al*, 2010; Verkerk *et al*, 2015) and elaborated in the Appendix.

## Statistics

Experiments with hiPSC-CMs were performed on cells obtained from $n \geq 3$ independent differentiations. Statistical analysis was carried out with SigmaStat 3.5 software, and data are presented as mean ± standard error of the mean. Normality and equal variance assumptions were tested with the Kolmogorov–Smirnov and the Levene median test, respectively. Groups were compared using one-way analysis of variance (ANOVA) followed by pairwise comparison using the Student–Newman–Keuls test or, in cases of failed normality and/or equal variance test, ANOVA based on ranks

## The paper explained

### Problem
Inherited arrhythmogenic disorders are one of the leading causes of sudden cardiac death (SCD). However, 30% of SCD cases are due to unknown gene mutations.

### Results
Here, we presented the clinical phenotype of patients from three different families with exercise-induced arrhythmias and identified mutations in the gene, *TECRL* by whole-exome sequencing. We generated human induced pluripotent stem cells (hiPSCs) from an affected patient and differentiated these to cardiomyocytes (CMs). We uncovered electrophysiological and calcium-handling abnormalities in CMs with a *TECRL* mutation compared to hiPSC-CMs generated from an unaffected family member. Moreover, mutant cells were prone to an increase in triggered electrical activity upon catecholaminergic stimulation with noradrenaline. This could be partially rescued by treatment with a class Ic antiarrhythmic drug, flecainide.

### Impact
Our study demonstrated that *TECRL* mutations are associated with a complex clinical phenotype with characteristics of both long QT syndrome (LQTS) and catecholaminergic polymorphic ventricular tachycardia (CPVT). Screening for mutations in *TECRL* should be implemented in symptomatic patient cohorts negative for mutations in classical LQTS and CPVT genes. This study also reiterates the importance of hiPSC models, especially valuable to assess the effect of novel gene variants on disease phenotype. In summary, findings presented in this study have major implications for improving diagnosis and management of inherited arrhythmia syndromes.

(Kruskal–Wallis test) followed by Dunn's test. $P < 0.05$ was considered statistically significant.

**Expanded View** for this article is available online.

## Acknowledgements

Funding for this study from the following sources is gratefully acknowledged: the Netherlands Organization for Health Research and Development (ZonMw-TOP 40-00812-98-12086) to H.D.D and (ZonMw-MKMD-40-42600-98-036) to R.P; European Research Council advanced grant (STEMCARDIOVASC-323182) to C.L.M; the Leenaards Foundation, the Swiss Institute of Bioinformatics, and the Swiss National Science Foundation (31003A-143914, 51RTP0_151019) grants to Z.K; Fondation Suisse de Cardiologie (No. 29283) to Z.A.B; the Netherlands CardioVascular Research Initiative, the Dutch Heart Foundation, Dutch Federation of University Medical Centers, the Netherlands Organization for Health Research and Development, and the Royal Netherlands Academy of Sciences (CVON-PREDICT) to A.A.M.W; Fondation de l'Institut de cardiologie de Montréal to JDR; and the Phillipa and Marvin Carsley Chair of Medicine of the University of Montreal to MT and RT. The authors thank the following people for contributing to the research presented in this manuscript: Laura Robb and Dr Blandine Mondésert of the Cardiovascular Genetics Center of the Montreal Heart Institute for their assistance in the characterization of the French Canadian probands and family members; Prof. Jacques S. Beckmann for initiating the next-generation sequence study of the Sudanese family and incorporating this family for exome sequencing; Dr Tom van Wezel (Department of Pathology, LUMC) for helpful discussions; M. Ohtaka, K. Nishimura, and M. Nakanishi (National Institute of Advanced Industrial Science and Technology, Japan) for providing the Sendai virus vectors; S. van de Pas and A. 't Jong from the LUMC hiPSC core facility for support with reprogramming; D. de Jong, K. Szuhai, and H. Tanke (Department of Molecular Cell Biology, LUMC) for karyotyping, Verena Schwach (Department of Anatomy & Embryology, LUMC) for support with cell culture, and Annemarie Kip (Department of Cardiology, LUMC) for help with the lentiviral vector production. The computations were performed at the Vital-IT Center (http://www.vital-it.ch) for high-performance computing of the SIB Swiss Institute of Bioinformatics. The authors are especially grateful to the patient families for tissue samples and their consent for the study.

## Author contributions

LA-G and EHA enrolled the Sudanese patient family into the study with necessary approvals and collected clinical data and blood samples/biopsies; MT, LR, and M-AC enrolled the French Canadian patient families into the study with necessary approvals and collected clinical data and blood samples; JS, MT, M-AC, RT, and AAMW analyzed the clinical data; ZAB initiated whole-exome sequencing of the Sudanese family members, carried out validation of identified variants by Sanger sequencing and coordinated the study; HJ performed exome sequencing; HJ, ZK, BS, and ALB analyzed exome sequencing data; JDR and MT initiated whole-exome sequencing of French Canadian samples; AA performed exome sequencing; M-AC, FL, and PG analyzed exome sequencing data; PG and AA carried out validation of identified variants by Sequenom Mass Array genotyping; AA and RG performed experiments; PG and RG performed data analysis and interpretation; HDD and RP designed the hiPSC research. HDD and CF generated the hiPSC lines; HDD, AB, TZ, GK, JJM-K, MH, and AOV performed experiments with the hiPSC lines; HDD, AB, GK, JJM-K, AOV, and RP performed analysis and interpretation of data from hiPSC lines; AB performed *TECRL* expression studies in mouse and sub-cellular localization experiments in cell lines; AAFV produced the lentiviral vectors for the *TECRL* knockdown studies; HDD, AOV, RP, RT, RG, and JDR wrote the manuscript with contributions from all the authors. CLM provided input on the design of the study and edited the manuscript. All authors read, provided input, and approved the manuscript.

## Conflict of interest

The authors declare that they have no conflict of interest.

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
