## [Review Process File · EMBO Molecular Medicine]

TECRL, a new life-threatening inherited arrhythmia gene associated with overlapping clinical features of both LQTS and CPVT

Harsha D Devalla, Roselle Gélinas, Elhadi H Aburawi, Abdelaziz Beqqali, Philippe Goyette, Christian Freund, Marie-A Chaix, Rafik Tadros, Hui Jiang, Antony Le Béchech, Jantine Monshouwer-Kloots, Tom Zwetsloot, Georgios Kosmidis, Frédéric Latour, Azadeh Alikashani, Maaïke Hoekstra, Jurg Schlaepfer, Christine L Mummery, Brian Stevenson, Zoltan Kutalik, Antoine AF de Vries, Léna Rivard, Arthur AM Wilde, Mario Talajic, Arie O Verkerk, Lihadh Al-Gazali, John D Rioux, Zahurul A Bhuiyan, Robert Passier

Corresponding author: Harsha Devalla & Robert Passier, Leiden University Medical Center
Zahurul Bhuiyan, Universitaire Vaudois (CHUV), Lausanne
Jean Rioux, Montreal Heart Institute

Review timeline:

Submission date:	21 August 2015
Editorial Decision:	13 September 2015
Revision received:	11 January 2016
Editorial Decision:	26 January 2016
Revision received:	27 July 2016
Editorial Decision:	18 August 2016
Revision received:	20 September 2016
Accepted:	23 September 2016

Transaction Report:

Editor: Roberto Buccione

1st Editorial Decision

13 September 2015

Thank you for the submission of your manuscript to EMBO Molecular Medicine.

In this case we experienced unusual difficulties in securing three willing and appropriate reviewers, also due to the overlap with the vacation period. As a further delay cannot be justified I have decided to proceed based on the two available consistent evaluations.

Both Reviewers are quite positive on your manuscript although they raise some issues that require your action. I will not dwell into much detail as their comments are detailed. I would like, however, to highlight a few main points.

Reviewer 1, as you will see, lists a few of issues including that s/he suggests undertaking further experimentation to better define the role of TECRL, for instance by analysing the phenotype of "control" iPS cells with knock down of TECRL. We agree that to address this and the other points would significantly enhance the significance of the study.

Reviewer 2 also suggests a number of actions to improve the manuscript. These include better statistical analysis (to this effect please see below on our checklist), streamlining the clinical details on the patients and others. Regarding the latter point, we agree and perhaps you might want to move some information to the supplementary information section. Reviewer 2, however, would also like you to experimentally address why you observed no noradrenaline-triggered delayed after depolarizations and whether this was due to the presence of the calcium indicator.

In conclusion, while publication of the paper cannot be considered at this stage, we would be pleased to consider a substantially revised submission, with the understanding that the Reviewers' concerns must be addressed with additional experimental data where appropriate and that acceptance of the manuscript will entail a second round of review.

Please note that it is EMBO Molecular Medicine policy to allow a single round of revision only and that, therefore, acceptance or rejection of the manuscript will depend on the completeness of your responses included in the next, final version of the manuscript.

As you might know, EMBO Molecular Medicine has a "scooping protection" policy, whereby similar findings that are published by others during review or revision are not a criterion for rejection. However, I do ask you to get in touch with us after three months if you have not completed your revision, to update us on the status. Please also contact us as soon as possible if similar work is published elsewhere.

Please note that EMBO Molecular Medicine now requires a complete author checklist (<http://embomolmed.embopress.org/authorguide#editorial3>) to be submitted with all revised manuscripts. Provision of the author checklist is mandatory at revision stage; The checklist is designed to enhance and standardize reporting of key information in research papers and to support reanalysis and repetition of experiments by the community. The list covers key information for figure panels and captions and focuses on statistics, the reporting of reagents, animal models and human subject-derived data, as well as guidance to optimise data accessibility.

I also suggest that you carefully adhere to our guidelines for publication in your next version, including presentation of statistical analyses (see also below) and our new requirements for supplemental data (Expanded View; <http://embomolmed.embopress.org/authorguide#expandedview>) to speed up the pre-acceptance process.

I look forward to seeing a revised form of your manuscript as soon as possible.

***** Reviewer's comments *****

Referee #1 (Remarks):

Devalla and colleagues generated iPSC from patients affected by a clinical CPVT syndrome to phenocopy this disease in the dish. Mutations of Ryr2 (observed in CPVT1) and Casq2 (observed in CPVT2) were not identified. Whole genome sequencing revealed a deletion of exon 3 in the *Tecrl* gene. Association of *Tecrl* with the ER and strong expression in the developing mouse heart were demonstrated. Diastolic calcium overload potentially as a consequence of delayed cytosolic calcium clearance was observed and suggested to be the underlying cause of DAD-like electrophysiological events. Noradrenaline enhanced DADs as anticipated for cardiomyocytes from patients with CPVT.

Major concerns:

- 1) Additional experiments are needed to underpin the mechanistic role of *Tecrl* in the observed phenotype. This would also strengthen the rationale for including *Tecrl* in genetic panel for CPVT screens. Would a knock-down of *Tecrl* in a control ESC or iPSC line lead to a similar phenotype? Alternatively, would overexpression of exon 3 deleted *Tecrl* induce a similar phenotype?
- 2) Data on *Tecrl* transcript and/or protein expression would be helpful to understand whether the

exon 3 deleted transcript/protein is present, absent, or compensated in cardiomyocytes by the non-affected allele in the TecrlHet cardiomyocytes.

3) Data on RyR2, Casq2, SERCA, PLB, NCX, LTCC protein abundance and phosphorylation should be provided to understand whether canonical calcium handling proteins are affected.

Minor concerns:

1) It should be noted in the discussion that the AP duration, independent of the condition, remains relatively short for a ventricular myocyte (APD₉₀: <200 ms). This does not limit the value of the study, but in my view provides further arguments of the use of iPSC in disease modelling despite an apparently immature electrophysiological phenotype.

Referee #2 (Comments on Novelty/Model System):

This is a very thorough study and of significant interest. The high ranking for technical quality is justified for all but one minor omission in the calcium handling studies (see comments to authors. In terms of medical impact - I have ranked it medium rather than high, simply because this is likely to be extremely rare - so whilst it has significant scientific interest I doubt that very many people will come across this condition.

Referee #2 (Remarks):

In this study, Devalla and colleagues have elucidated the genetic basis of a rare but highly malignant form of CPVT. They have identified a new gene involved in inherited arrhythmia syndromes, TECRL, and largely determined the underlying molecular basis of the pathology in this family. The approach they have taken is very logical and straight-forward, but that should not underestimate the magnitude of what they have achieved in terms of drilling down into the underlying mechanism. I have no major concerns with the study but a few points that should be clarified.

1. In the abstract I would include a comment about the slower rise time of the calcium transients and the lower SR Calcium stores as evidenced by the decreased magnitude of caffeine-induced transients.
2. p6. I do not think that it is necessary to add in so much of the clinical data. Of most relevance to this study are the electrical phenotypes - I do not think the readers need to know all the agonising details of the patients' stays in ICU.
3. p14. The authors comment that a raised diastolic [Ca²⁺]_i would predispose to DADs. However, a decreased NCX activity could reduce DADs - the authors should add a comment along these lines - and modify their conclusions in the discussion along the lines of despite the reduced NCX, the increase in diastolic calcium was sufficient to still result in DADs.
4. For me, the only slight weakness in the study was the lack of data for calcium transients in the presence of NA. It is possible that the extra buffering of [Ca²⁺]_i in the presence of indo-I might abrogate any NA-triggered DADs. If the authors have tried these experiments and did not observe DADs - then they should say so and provide a possible explanation (such as buffering effects). If they have not tried them then I would strongly encourage them to do these experiments as they will be very interesting, particularly if they parallel the findings with the AP recordings.
5. p28. Statistics. It is well known that there is considerable variability in AP and calcium transient characteristics in hiPS CMs from one differentiation to the next and even from one plating to the next of the same cell lines. It is therefore very important that the authors have repeated measurements on cells derived from at least three separate differentiations. For completeness, can the authors include in the relevant figure legends details of how many cells from how many independent platings are included. For example in Figure 4, rather than reporting n=15-18 it would be better to state n=4-6 from 3 different platings/differentiations with total of n=15-18 (or whatever the specific numbers are).
6. Figure 8: If y axis is percentage then I presume the values on these should be 0, 50, 100 and 150 (not 0, 0.5, 1.0, 1.5).

Referee #1 (Remarks):

Devalla and colleagues generated iPSC from patients affected by a clinical CPVT syndrome to phenocopy this disease in the dish. Mutations of RyR2 (observed in CPVT1) and Casq2 (observed in CPVT2) were not identified. Whole genome sequencing revealed a deletion of exon 3 in the Tecrl gene. Association of Tecrl with the ER and strong expression in the developing mouse heart were demonstrated. Diastolic calcium overload potentially as a consequence of delayed cytosolic calcium clearance was observed and suggested to be the underlying cause of DAD-like electrophysiological events. Noradrenaline enhanced DADs as anticipated for cardiomyocytes from patients with CPVT.

Major concerns:

1) Additional experiments are needed to underpin the mechanistic role of Tecrl in the observed phenotype. This would also strengthen the rationale for including Tecrl in genetic panel for CPVT screens. Would a knock-down of Tecrl in a control ESC or iPSC line lead to a similar phenotype? Alternatively, would overexpression of exon 3 deleted Tecrl induce a similar phenotype?

We thank the reviewer for this question. As suggested, we performed shRNA-mediated knockdown of *TECRL* in control hESC-CMs, which resulted in a significant prolongation of action potential duration (APD) at 20%, 50% and 90% of repolarization (APD₂₀, APD₅₀ and APD₉₀) compared with CMs treated with a scrambled shRNA (Fig. S5). This is in agreement with our findings in patient-derived hiPSC-CMs with a *TECRLc.331+1G>A* mutation, which also displayed significantly prolonged action potentials at 20% repolarization (APD₂₀) and there was a clear trend for an increase in APD at 50% (APD₅₀) and 90% (APD₉₀) repolarization compared to CTRL-hiPSC-CMs (Fig. 6C). Furthermore, hESC-CMs with *TECRL* knockdown showed decreased RYR2 (by 56%) and CASQ2 (by 18%) protein levels (Fig. S5E) similar to hiPSC-CMs with *TECRLc.331+1G>A* mutation (Fig. S4). These results suggest a link between *TECRL* and calcium homeostasis in CMs. Additionally, we also tried to evaluate the susceptibility of hESC-CMs with *TECRL* knockdown, to triggered activity. As done with hiPSC-CMs, we used a fast pacing episode (3-Hz; 10-seconds), followed by a 10-second pause in the absence or presence of 10 nM Noradrenaline (NA). However, due to extremely long APs of *TECRL* knockdown CMs, most cells were not able to follow a fast-pacing protocol and therefore, we could not provide this data in the manuscript.

We have included the results from *TECRL* knockdown experiments in the revised version of the manuscript (Results: Page 14-16 of main manuscript; Figure: S5 of supplementary information).

2) Data on Tecrl transcript and/or protein expression would be helpful to understand whether the exon 3 deleted transcript/protein is present, absent, or compensated in CMs by the non-affected allele in the TecrlHet cardiomyocytes.

In *TECRL*_{Het}-hiPSC-CMs, we observed the presence of both *TECRL* transcripts by RT-PCR with primers designed to target exons 2-4 of *TECRL*: a longer 171bp product containing exon3 and a shorter 126 bp product lacking exon3 (Fig. 3G). So, it appears that the wild-type transcript does not compensate for the mutated transcript in *TECRL*_{Het}-hiPSC-CMs.

We also attempted to detect TECRL protein on western blot and immunostainings using a TECRL specific antibody. Unfortunately, both commercially available and custom-made antibodies were unsuccessful in detecting TECRL protein.

3) Data on RyR2, Casq2, SERCA, PLB, NCX, LTCC protein abundance and phosphorylation should be provided to understand whether canonical calcium handling proteins are affected.

We thank the reviewer for this question and these experiments have been very informative. By western blot, we assessed protein expression of several calcium handling proteins in hiPSC-CMs as well as in TECRL-knockdown CMs. RYR2 (by 52%) and CASQ2 (by 85%) were found to be consistently downregulated in TECRL_{Hom}-hiPSC-CMs and representative blots are shown in Fig. S4. However, we did not observe any differences in the protein levels of SERCA, PLB, NCX or CA_v1.2 between CTRL and TECRL_{Hom}-hiPSC-CMs. Similarly, we also observed a reduction in the protein expression of RYR2 (by 56%) and CASQ2 (by 18%) in TECRL-knockdown CMs (Fig. S5E). These results suggest that the *c.331+1G>A* mutation or a deficiency in TECRL influences the expression of RYR2 and CASQ2, which have both been implicated in CPVT.

However, WB to detect phosphorylated RYR2 or phosphorylated PLB were not successful likely due to relatively low expression of these proteins in iPSC- and hESC-derived CMs. Another explanation of the lack of phospho-specific signal could be that the phosphatase inhibition by NaF and Na₃VO₄ was not sufficient. To our knowledge, there has not been any published data on detection of phosphorylated calcium handling proteins in hPSC-derived CMs.

We have included the results from WB of canonical calcium handling proteins in the revised version of the manuscript (Results: Page-12; 15-16 of main manuscript; Figure: S4; S5E of supplementary information).

Minor concerns:

1) It should be noted in the discussion that the AP duration, independent of the condition, remains relatively short for a ventricular myocyte (APD₉₀: <200 ms). This does not limit the value of the study, but in my view provides further arguments of the use of iPSC in disease modelling despite an apparently immature electrophysiological phenotype.

The reviewer is right that the APD in PSC-derived CMs is shorter than in freshly isolated adult ventricular CMs. We have mentioned this point in the discussion (Page-22 of main manuscript) and that this does not preclude their use in revealing disease phenotypes

Referee #2 (Comments on Novelty/Model System):

This is a very thorough study and of significant interest. The high ranking for technical quality is justified for all but one minor omission in the calcium handling studies (see comments to authors. In terms of medical impact - I have ranked it medium rather than high, simply because this is likely to be extremely rare - so whilst it has significant scientific interest I doubt that very many people will come across this condition.

Referee #2 (Remarks):

In this study, Devalla and colleagues have elucidated the genetic basis of a rare but highly malignant form of CPVT. They have identified a new gene involved in inherited arrhythmia syndromes, TECRL, and largely determined the underlying molecular basis of the pathology in this family. The approach they have taken is very logical and straight-forward, but that should not underestimate the magnitude of what they have achieved in terms of drilling down into the underlying mechanism. I have no major concerns with the study but a few points that should be clarified.

1. In the abstract I would include a comment about the slower rise time of the calcium

transients and the lower SR Calcium stores as evidenced by the decreased magnitude of caffeine-induced transients.

We thank the reviewer for this suggestion and have now included this information in the abstract (Page-3 of the main manuscript).

2. p6. I do not think that it is necessary to add in so much of the clinical data. Of most relevance to this study are the electrical phenotypes - I do not think the readers need to know all the agonising details of the patients' stays in ICU.

As suggested by the reviewer and the editors, we have moved most of the clinical data to the supplementary section. Brief description of subject IV:13 (also investigated by hiPSCs in this study) and a table summarizing the other patients in the family have been left in the main manuscript.

3. p14. The authors comment that a raised diastolic $[Ca^{2+}]_i$ would predispose to DADs. However, a decreased NCX activity could reduce DADs - the authors should add a comment along these lines - and modify their conclusions in the discussion along the lines of despite the reduced NCX, the increase in diastolic calcium was sufficient to still result in DADs.

The reviewer raises a valid point and we have now included this in the discussion (Page-20 of main manuscript).

4. For me, the only slight weakness in the study was the lack of data for calcium transients in the presence of NA. It is possible that the extra buffering of $[Ca^{2+}]_i$ in the presence of indo-1 might abrogate any NA-triggered DADs. If the authors have tried these experiments and did not observe DADs - then they should say so and provide a possible explanation (such as buffering effects). If they have not tried them then I would strongly encourage them to do these experiments as they will be very interesting, particularly if they parallel the findings with the AP recordings.

The reviewer raises an important point, which we also recognized during our study. We tried extensively to measure spontaneous Ca^{2+} -release in hiPSC-CMs in response to NA, with a protocol similar to our patch clamp experiments. Our intention was therefore, to use a fast pacing episode (3-Hz; 10-seconds), followed by a 10-second pause in the absence or presence of 10 nM Noradrenaline (NA). Unfortunately we were not able to pace hiPSC-CMs with field stimulation at 3 Hz and had to discontinued this approach. It is a general observation that it is more difficult to pace CMs at fast frequencies with field stimulation. The reviewer is also right that extra buffering of $[Ca^{2+}]_i$ may occur in the presence of Indo-I. We also observed that hiPSC-CMs become quiescent after Indo-1 loading, as also mentioned in the methods section (page-6 of supplementary information) and this additionally might have hampered the measurements.

5. p28. Statistics. It is well known that there is considerable variability in AP and calcium transient characteristics in hiPSC CMs from one differentiation to the next and even from one plating to the next of the same cell lines. It is therefore very important that the authors have repeated measurements on cells derived from at least three separate differentiations. For completeness, can the authors include in the relevant figure legends details of how many cells from how many independent platings are included. For example in Figure 4, rather than reporting $n=15-18$ it would be better to state $n=4-6$ from 3 different platings/differentiations with total of $n=15-18$ (or whatever the specific numbers are).

As the reviewer rightly points out, variation in electrical and functional parameters can be noted between differentiation to differentiation from hiPSC-lines. Data presented in this study for all of the experiments (electrophysiology, calcium measurements and WBs included in the revised manuscript) were acquired from at least three independent differentiations. For clarity, this is also mentioned in the subsection 'statistics' of the methods in the main manuscript (Page-25 of main manuscript). As suggested by the reviewer, figure legends for individual figures also now contain additional information about the number of cells obtained per experiment.

6. *Figure 8: If y axis is percentage then I presume the values on these should be 0, 50, 100 and 150 (not 0, 0.5, 1.0, 1.5).*

We thank the reviewer for bringing this to our notice. The values on y-axis in Fig.8B and 8D are indeed a percentage and have now been corrected

2nd Editorial Decision

26 January 2016

Thank you for the submission of your revised manuscript to EMBO Molecular Medicine. We have now received the enclosed reports from the referees that were asked to re-assess it. As you will see the reviewers are now globally supportive and I am pleased to inform you that we will be able to accept your manuscript pending the following final amendments:

- 1) Please comply with Reviewer 2's final request for a minor clarification
- 2) The manuscript is still missing "The paper explained" section". EMBO Molecular Medicine articles are accompanied by a summary of the articles to emphasize the major findings in the paper and their medical implications for the non-specialist reader. Please provide a draft summary of your article highlighting the medical issue you are addressing, the results obtained and their clinical impact. Please refer to any of our published papers as a reference (embomolmed.org). This may be edited to ensure that readers understand the significance and context of the research. Please refer to any of our published articles for an example.
- 3) As per our Author Guidelines, the description of all reported data that includes statistical testing must state the name of the statistical test used to generate error bars and P values, the number (n) of independent experiments underlying each data point (not replicate measures of one sample), and the actual P value for each test (not merely 'significant' or 'P < 0.05').
- 4) We are now encouraging the publication of source data, particularly for electrophoretic gels and blots, with the aim of making primary data more accessible and transparent to the reader. Would you be willing to provide a PDF file per figure that contains the original, uncropped and unprocessed scans of all or at least the key gels used in the manuscript? The PDF files should be labeled with the appropriate figure/panel number, and should have molecular weight markers; further annotation may be useful but is not essential. The PDF files will be published online with the article as supplementary "Source Data" files. If you have any questions regarding this just contact me.
- 5) Every published paper includes a 'Synopsis' to further enhance discoverability. Synopses are displayed on the journal webpage and are freely accessible to all readers. They include a short standfirst as well as 2-5 one sentence bullet points that summarise the paper. Please provide the synopsis including the short list of bullet points that summarise the key NEW findings. The bullet points should be designed to be complementary to the abstract - i.e. not repeat the same text. We encourage inclusion of key acronyms and quantitative information. Please use the passive voice. Please attach this information in a separate file or send them by email, we will incorporate it accordingly. You are also welcome to suggest a striking image or visual abstract to illustrate your article. If you do please provide a jpeg file 550 px-wide x 400-px high.
- 6) As per our guidelines for Authors, the figure call-outs within the article and inside "suppl. information" need to be adjusted to Appendix Figure S1, Appendix Figure S2, etc ... and the supplementary information file must be renamed "Appendix".
- 7) Please note that we now mandate that all corresponding authors list an ORCID digital identifier. You may do so through our web platform upon submission and the procedure takes <90 seconds to complete. We encourage all authors to supply an ORCID identifier, which will be linked to their name for unambiguous name identification.

Please submit your revised manuscript within two weeks. I look forward to seeing a revised form of your manuscript as soon as possible.

***** Reviewer's comments *****

Referee #1 (Remarks):

The authors have addressed my critiques.

Referee #2 (Remarks):

The authors have addressed all of the concerns I raised. The one very minor point I would like to address - is that they indicate in the main manuscript that they did try to measure spontaneous release events in response to NA (to complement the electrical measurements) but were not able to do so due to the technical limitations of using field stimulation (rather than direct pacing) (as indicated in their response to my previous point 4). This could either be mentioned at the end of the relevant results section (p16) or in the discussion.

2nd Revision - authors' response

27 July 2016

On behalf of all the authors, we thank you for the feedback on earlier versions of our manuscript titled "A splice site mutation in *TECRL* is associated with inherited lethal arrhythmias in humans". Based on your comments, we had revised the article in January this year that was then accepted.

However, during the evaluation of the previous version, we learnt that the group of John Rioux from the University of Montreal in Canada also identified patients who carried a rare missense homozygous mutation in *TECRL* and a clinical phenotype of LQTS but also adrenergic-induced arrhythmias. In the Sudanese family we had presented in our study earlier, we had observed adrenergic-induced arrhythmias in patients with a splice-site mutation in *TECRL* who were clinically diagnosed with CPVT but QTc prolongation was also noted in some patients.

This confirmed our hypothesis that indeed *TECRL* is an import gene associated with stress-induced arrhythmias (with overlapping features of both LQTS and CPVT, also evident in patient-derived hiPSC-cardiomyocytes and knockdown experiments). We felt that adding clinical data from unrelated patients would maximize the clinical impact with the message that genetic testing for mutations in *TECRL* should be implemented in patients negative for mutations in classic LQTS and CPVT genes. Also, we wanted to present the story as best as possible with all of the information known to us at this point and hence took the decision of adding the clinical data from Canada to our existing data.

We apologize for the delay in resubmission of the combined manuscript, which was due to collecting additional patient data and organizing the manuscript but we are now confident that the message of the study is stronger with the inclusion of multiple unrelated patients. Findings presented in our manuscript add to the spectrum of LQTS and CPVT associated genes that can be implemented in diagnostic screenings. Our work also reiterates the value of hiPSC derivatives to study human disorders, rare diseases in particular. The revised manuscript is now titled "*TECRL*, a new lifethreatening inherited arrhythmia gene associated with overlapping clinical features of both LQTS and CPVT".

In this new version, Figure-1 has been modified to present clinical features of three different patients. In Figure-2, we have replaced the panel showing expression of *TECRL* in adult mouse tissues with expression of *TECRL* in human tissues. The other figures and results remain unchanged. Text in parts of the manuscript has been adapted to fit the new information in the revised version.

We thank you for your time and look forward to your input.

Thank you for the submission of your revised manuscript to EMBO Molecular Medicine. We have now received the enclosed report from reviewer 1 who was asked to re-assess it also on behalf of reviewer 2, who was not available.

As you will see the reviewer is satisfied with your amended and integrated manuscript (but please note his/her remark on figure labeling) and I am pleased to inform you that we will be able to accept your manuscript pending the following final amendments. In fact, although I had mentioned many of them in my previous decision letter, I note that they have not yet been dealt with:

1) Please update the author list in the manuscript submission interface when you upload your revised manuscript

2) The manuscript is still missing "The paper explained" section". EMBO Molecular Medicine articles are accompanied by a summary of the articles to emphasize the major findings in the paper and their medical implications for the non-specialist reader. Please provide a draft summary of your article highlighting the medical issue you are addressing, the results obtained and their clinical impact. Please refer to any of our published papers as a reference (embomolmed.org). This may be edited to ensure that readers understand the significance and context of the research. Please refer to any of our published articles for an example.

3) As per our Author Guidelines, the description of all reported data that includes statistical testing must state the name of the statistical test used to generate error bars and P values, the number (n) of independent experiments underlying each data point (not replicate measures of one sample), and the ACTUAL P value for each test (not merely 'significant' or ' $P < 0.05$ ').

4) We are now encouraging the publication of source data, particularly for electrophoretic gels and blots, with the aim of making primary data more accessible and transparent to the reader. Would you be willing to provide a PDF file per figure that contains the original, uncropped and unprocessed scans of all or at least the key gels used in the manuscript? The PDF files should be labeled with the appropriate figure/panel number, and should have molecular weight markers; further annotation may be useful but is not essential. The PDF files will be published online with the article as supplementary "Source Data" files. If you have any questions regarding this just contact me.

5) Every published paper includes a 'Synopsis' to further enhance discoverability. Synopses are displayed on the journal webpage and are freely accessible to all readers. They include a short standfirst as well as 2-5 one sentence bullet points that summarise the paper. Please provide the synopsis including the short list of bullet points that summarise the key NEW findings. The bullet points should be designed to be complementary to the abstract - i.e. not repeat the same text. We encourage inclusion of key acronyms and quantitative information. Please use the passive voice. Please attach this information in a separate file or send them by email, we will incorporate it accordingly. You are also welcome to suggest a striking image or visual abstract to illustrate your article. If you do please provide a jpeg file 550 px-wide x 400-px high.

6) Please note that EMBO Molecular Medicine now requires a complete author checklist (<http://embomolmed.embopress.org/authorguide#editorial3>) to be submitted with all revised manuscripts. Provision of the author checklist is mandatory at revision stage; The checklist is designed to enhance and standardize reporting of key information in research papers and to support reanalysis and repetition of experiments by the community. The list covers key information for figure panels and captions and focuses on statistics, the reporting of reagents, animal models and human subject-derived data, as well as guidance to optimise data accessibility. The Author checklist will be published alongside the paper, in case of acceptance, within the transparent review process file.

I look forward to seeing a revised form of your manuscript as soon as possible, and in any case within two weeks, so that we can rapidly proceed with formal acceptance and production.

***** Reviewer's comments *****

Referee #1 (Remarks):

The additional clinical and experimental data from Montreal strongly supports the hypothesis that TECRL mutations are associated with ventricular arrhythmia. The authors must be congratulated on their efforts to further strengthen the translational relevance of their work.

On page 15 Fig 3H should be 3G.

Corresponding Author Name: H.D.Devalla
Manuscript Number: EMM-2015-05719-V3